# Analyzing Inverse Problems with Invertible Neural Networks

**Lynton Ardizzone[1], Jakob Kruse[1], Sebastian Wirkert[2],**
**Daniel Rahner[3], Eric W. Pellegrini[3], Ralf S. Klessen[3],**
**Lena Maier-Hein[2], Carsten Rother[1], Ullrich Köthe[1]**
[1]Visual Learning Lab Heidelberg, [2]German Cancer Research Center (DKFZ),
[3]Zentrum für Astronomie der Universität Heidelberg (ZAH)
[1]lynton.ardizzone@iwr.uni-heidelberg.de,
[2]s.wirkert@dkfz-heidelberg.de, [3]daniel.rahner@uni-heidelberg.de

## Abstract

For many applications, in particular in natural science, the task is to determine hidden system parameters from a set of measurements. Often, the forward process from parameter- to measurement-space is well-defined, whereas the inverse problem is ambiguous: multiple parameter sets can result in the same measurement. To fully characterize this ambiguity, the full posterior parameter *distribution*, conditioned on an observed measurement, has to be determined. We argue that a particular class of neural networks is well suited for this task – so-called Invertible Neural Networks (INNs). Unlike classical neural networks, which attempt to solve the ambiguous inverse problem directly, INNs focus on learning the forward process, using additional latent output variables to capture the information otherwise lost. Due to invertibility, a model of the corresponding inverse process is learned implicitly. Given a specific measurement and the distribution of the latent variables, the inverse pass of the INN provides the full posterior over parameter space. We prove theoretically and verify experimentally, on artificial data and real-world problems from medicine and astrophysics, that INNs are a powerful analysis tool to find multi-modalities in parameter space, uncover parameter correlations, and identify unrecoverable parameters.

## 1 Introduction

When analyzing complex physical systems, a common problem is that the system parameters of interest cannot be measured directly. For many of these systems, scientists have developed sophisticated theories on how measurable quantities $\mathbf{y}$ arise from the hidden parameters $\mathbf{x}$. We will call such mappings the *forward process*. However, the *inverse* process is required to infer the hidden states of a system from measurements. Unfortunately, the inverse is often both intractable and ill-posed, since crucial information is lost in the forward process.

To fully assess the diversity of possible inverse solutions for a given measurement, an inverse solver should be able to estimate the *complete posterior* of the parameters, conditioned on an observation. This makes it possible to quantify uncertainty, reveal multi-modal distributions, and identify degenerate and unrecoverable parameters – all highly relevant for applications in natural science. In this paper, we ask if *invertible neural networks* (INNs) are a suitable model class for this task. INNs are characterized by three properties:

(i) The mapping from inputs to outputs is bijective, i.e. its inverse exists,

(ii) both forward and inverse mapping are efficiently computable, and

(iii) both mappings have a tractable Jacobian, which allows explicit computation of posterior probabilities.

Networks that are invertible by construction offer a unique opportunity: We can train them on the well-understood forward process $\mathbf{x} \to \mathbf{y}$ and get the inverse $\mathbf{y} \to \mathbf{x}$ for free by

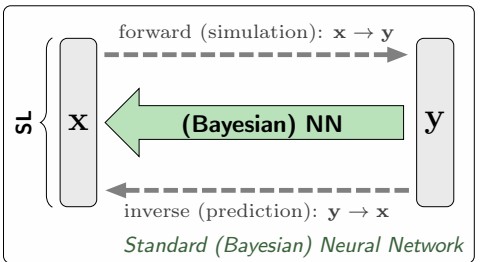 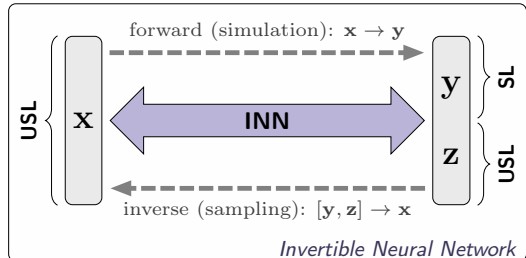

Figure 1: **Abstract comparison of standard approach *(left)* and ours *(right)*.** The standard direct approach requires a discriminative, supervised loss (**SL**) term between predicted and true $\mathbf{x}$, causing problems when $\mathbf{y} \to \mathbf{x}$ is ambiguous. Our network uses a supervised loss only for the well-defined forward process $\mathbf{x} \to \mathbf{y}$. Generated $\mathbf{x}$ are required to follow the prior $p(\mathbf{x})$ by an unsupervised loss (**USL**), while the latent variables $\mathbf{z}$ are made to follow a Gaussian distribution, also by an unsupervised loss. See details in Section 3.3.

running them backwards at prediction time. To counteract the inherent information loss of the forward process, we introduce additional *latent* output variables $\mathbf{z}$, which capture the information about $\mathbf{x}$ that is *not* contained in $\mathbf{y}$. Thus, our INN learns to associate hidden parameter values $\mathbf{x}$ with unique pairs $[\mathbf{y}, \mathbf{z}]$ of measurements and latent variables. Forward training optimizes the mapping $[\mathbf{y}, \mathbf{z}] = f(\mathbf{x})$ and implicitly determines its inverse $\mathbf{x} = f^{-1}(\mathbf{y}, \mathbf{z}) = g(\mathbf{y}, \mathbf{z})$. Additionally, we make sure that the density $p(\mathbf{z})$ of the latent variables is shaped as a Gaussian distribution. Thus, the INN represents the desired posterior $p(\mathbf{x} \,|\, \mathbf{y})$ by a deterministic function $\mathbf{x} = g(\mathbf{y}, \mathbf{z})$ that transforms ("pushes") the known distribution $p(\mathbf{z})$ to $\mathbf{x}$-space, conditional on $\mathbf{y}$.

Compared to standard approaches (see Fig. 1, *left*), INNs circumvent a fundamental difficulty of learning inverse problems: Defining a sensible supervised loss for direct posterior learning is problematic since it requires prior knowledge about that posterior's behavior, constituting a kind of hen-end-egg problem. If the loss does not match the possibly complicated (e.g. multimodal) shape of the posterior, learning will converge to incorrect or misleading solutions. Since the forward process is usually much simpler and better understood, forward training diminishes this difficulty. Specifically, we make the following contributions:

- We show that the full posterior of an inverse problem can be estimated with invertible networks, both theoretically in the asymptotic limit of zero loss, and practically on synthetic and real-world data from astrophysics and medicine.

- The architectural restrictions imposed by invertibility do not seem to have detrimental effects on our network's representational power.

- While forward training is sufficient in the asymptotic limit, we find that a combination with *unsupervised backward training* improves results on finite training sets.

- In our applications, our approach to learning the posterior compares favourably to approximate Bayesian computation (ABC) and conditional VAEs. This enables identifying unrecoverable parameters, parameter correlations and multimodalities.

## 2 Related work

Modeling the conditional posterior of an inverse process is a classical statistical task that can in principle be solved by Bayesian methods. Unfortunately, exact Bayesian treatment of real-world problems is usually intractable. The most common (but expensive) solution is to resort to sampling, typically by a variant of Markov Chain Monte Carlo (Robert and Casella, 2004; Gamerman and Lopes, 2006). If a model $\mathbf{y} = s(\mathbf{x})$ for the forward process is available, approximate Bayesian computation (ABC) is often preferred, which embeds the forward model in a rejection sampling scheme for the posterior $p(\mathbf{x}|\mathbf{y})$ (Sunnåker et al., 2013; Lintusaari et al., 2017; Wilkinson, 2013).

Variational methods offer a more efficient alternative, approximating the posterior by an optimally chosen member of a tractable distribution family (Blei et al., 2017). Neural

networks can be trained to predict accurate sufficient statistics for parametric posteriors (Papamakarios and Murray, 2016; Siddharth et al., 2017), or can be designed to learn a mean-field distribution for the network's weights via dropout variational inference (Gal and Ghahramani, 2015; Kingma et al., 2015). Both ideas can be combined (Kendall and Gal, 2017) to differentiate between data-related and model-related uncertainty. However, the restriction to limited distribution families fails if the true distribution is too complex (e.g. when it requires multiple modes to represent ambiguous or degenerate solutions) and essentially counters the ability of neural networks to act as *universal* approximators. Conditional GANs (cGANs; Mirza and Osindero, 2014; Isola et al., 2017) overcome this restriction in principle, but often lack satisfactory diversity in practice (Zhu et al., 2017b). For our tasks, conditional variational autoencoders (cVAEs; Sohn et al., 2015) perform better than cGANs, and are also conceptually closer to our approach (see appendix Sec. 2), and hence serve as a baseline in our experiments.

Generative modeling via learning of a non-linear transformation between the data distribution and a simple prior distribution (Deco and Brauer, 1995; Hyvärinen and Pajunen, 1999) has the potential to solve these problems. Today, this approach is often formulated as a *normalizing flow* (Tabak et al., 2010; Tabak and Turner, 2013), which gradually transforms a normal density into the desired data density and relies on bijectivity to ensure the mapping's validity. These ideas were applied to neural networks by Deco and Brauer (1995); Rippel and Adams (2013); Rezende and Mohamed (2015) and refined by Tomczak and Welling (2016); Berg et al. (2018); Trippe and Turner (2018). Today, the most common realizations use *auto-regressive flows*, where the density is decomposed according to the Bayesian chain rule (Kingma et al., 2016; Huang et al., 2018; Germain et al., 2015; Papamakarios et al., 2017; Oord et al., 2016; Kolesnikov and Lampert, 2017; Salimans et al., 2017; Uria et al., 2016). These networks successfully learned unconditional generative distributions for artificial data and standard image sets (e.g. MNIST, CelebA, LSUN bedrooms), and some encouraging results for conditional modeling exist as well (Oord et al., 2016; Salimans et al., 2017; Papamakarios et al., 2017; Uria et al., 2016).

These normalizing flows possess property (i) of an INN, and are usually designed to fulfill requirement (iii) as well. In other words, flow-based networks are invertible in principle, but the actual computation of their inverse is too costly to be practical, i.e. INN property (ii) is not fulfilled. This precludes the possibility of bi-directional or cyclic training, which has been shown to be very beneficial in generative adversarial nets and auto-encoders (Zhu et al., 2017a; Dumoulin et al., 2016; Donahue et al., 2017; Teng et al., 2018). In fact, optimization for cycle consistency forces such models to converge to invertible architectures, making fully invertible networks a natural choice. True INNs can be built using *coupling layers*, as introduced in the NICE (Dinh et al., 2014) and RealNVP (Dinh et al., 2016) architectures. Despite their simple design and training, these networks were rarely studied: Gomez et al. (2017) used a NICE-like design as a memory-efficient alternative to residual networks, Jacobsen et al. (2018) demonstrated that the lack of information reduction from input to representation does not cause overfitting, and Schirrmeister et al. (2018) trained such a network as an adversarial autoencoder. Danihelka et al. (2017) showed that minimization of an adversarial loss is superior to maximum likelihood training in RealNVPs, whereas the Flow-GAN of Grover et al. (2017) performs even better using bidirectional training, a combination of maximum likelihood and adversarial loss. The Glow architecture by Kingma and Dhariwal (2018) incorporates invertible 1x1 convolutions into RealNVPs to achieve impressive image manipulations. This line of research inspired us to extend RealNVPs for the task of computing posteriors in real-world inverse problems from natural and life sciences.

## 3 Methods

### 3.1 Problem specification

We consider a common scenario in natural and life sciences: Researchers are interested in a set of variables $\mathbf{x} \in \mathbb{R}^D$ describing some phenomenon of interest, but only variables $\mathbf{y} \in \mathbb{R}^M$ can actually be observed, for which the theory of the respective research field provides a model $\mathbf{y} = s(\mathbf{x})$ for the forward process. Since the transformation from $\mathbf{x}$ to $\mathbf{y}$ incurs an

information loss, the intrinsic dimension $m$ of $\mathbf{y}$ is in general smaller than $D$, even if the nominal dimensions satisfy $M > D$. Hence we want to express the inverse model as a conditional probability $p(\mathbf{x}\,|\,\mathbf{y})$, but its mathematical derivation from the forward model is intractable in the applications we are going to address.

We aim at approximating $p(\mathbf{x}\,|\,\mathbf{y})$ by a tractable model $q(\mathbf{x}\,|\,\mathbf{y})$, taking advantage of the possibility to create an arbitrary amount of training data $\{(\mathbf{x}_i, \mathbf{y}_i)\}_{i=1}^N$ from the known forward model $s(\mathbf{x})$ and a suitable prior $p(\mathbf{x})$. While this would allow for training of a standard regression model, we want to approximate the full posterior probability. To this end, we introduce a latent random variable $\mathbf{z} \in \mathbb{R}^K$ drawn from a multi-variate standard normal distribution and reparametrize $q(\mathbf{x}\,|\,\mathbf{y})$ in terms of a deterministic function $g$ of $\mathbf{y}$ and $\mathbf{z}$, represented by a neural network with parameters $\theta$:

$$\mathbf{x} = g(\mathbf{y}, \mathbf{z}; \theta) \quad \text{with} \quad \mathbf{z} \sim p(\mathbf{z}) = \mathcal{N}(\mathbf{z}; 0, I_K). \tag{1}$$

Note that we distinguish between *hidden parameters* $\mathbf{x}$ representing unobservable real-world properties and *latent variables* $\mathbf{z}$ carrying information intrinsic to our model. Choosing a Gaussian prior for $\mathbf{z}$ poses no additional limitation, as proven by the theory of non-linear independent component analysis (Hyvärinen and Pajunen, 1999).

In contrast to standard methodology, we propose to learn the model $g(\mathbf{y}, \mathbf{z}; \theta)$ of the inverse process *jointly* with a model $f(\mathbf{x}; \theta)$ approximating the known forward process $s(\mathbf{x})$:

$$[\mathbf{y}, \mathbf{z}] = f(\mathbf{x}; \theta) = [f_{\mathbf{y}}(\mathbf{x}; \theta), f_{\mathbf{z}}(\mathbf{x}; \theta)] = g^{-1}(\mathbf{x}; \theta) \quad \text{with} \quad f_{\mathbf{y}}(\mathbf{x}; \theta) \approx s(\mathbf{x}). \tag{2}$$

Functions $f$ and $g$ share the same parameters $\theta$ and are implemented by a *single invertible neural network*. Our experiments show that joint *bi-directional training* of $f$ and $g$ avoids many complications arising in e.g. cVAEs or Bayesian neural networks, which have to learn the forward process implicitly.

The relation $f = g^{-1}$ is enforced by the invertible network architecture, provided that the nominal and intrinsic dimensions of both sides match. When $m \le M$ denotes the intrinsic dimension of $\mathbf{y}$, the latent variable $\mathbf{z}$ must have dimension $K = D - m$, assuming that the intrinsic dimension of $\mathbf{x}$ equals its nominal dimension $D$. If the resulting nominal output dimension $M + K$ exceeds $D$, we augment the input with a vector $\mathbf{x}_0 \in \mathbb{R}^{M+K-D}$ of zeros and replace $\mathbf{x}$ with the concatenation $[\mathbf{x}, \mathbf{x}_0]$ everywhere. Combining these definitions, our network expresses $q(\mathbf{x}\,|\,\mathbf{y})$ as

$$q\big(\mathbf{x} = g(\mathbf{y}, \mathbf{z}; \theta)\,|\,\mathbf{y}\big) = p(\mathbf{z})\,\big|J_{\mathbf{x}}\big|^{-1}, \qquad J_{\mathbf{x}} = \det\left(\left.\frac{\partial g(\mathbf{y}, \mathbf{z}; \theta)}{\partial[\mathbf{y}, \mathbf{z}]}\right|_{\mathbf{y}, f_{\mathbf{z}}(\mathbf{x})}\right) \tag{3}$$

with Jacobian determinant $J_{\mathbf{x}}$. When using coupling layers, according to Dinh et al. (2016), computation of $J_{\mathbf{x}}$ is simple, as each transformation has a triangular Jacobian matrix.

## 3.2 Invertible architecture

To create a fully invertible neural network, we follow the architecture proposed by Dinh et al. (2016): The basic unit of this network is a reversible block consisting of two complementary affine coupling layers. Hereby, the block's input vector $\mathbf{u}$ is split into two halves, $\mathbf{u}_1$ and $\mathbf{u}_2$, which are transformed by an affine function with coefficients $\exp(s_i)$ and $t_i$ ($i \in \{1, 2\}$), using element-wise multiplication ($\odot$) and addition:

$$\mathbf{v}_1 = \mathbf{u}_1 \odot \exp(s_2(\mathbf{u}_2)) + t_2(\mathbf{u}_2), \qquad \mathbf{v}_2 = \mathbf{u}_2 \odot \exp(s_1(\mathbf{v}_1)) + t_1(\mathbf{v}_1). \tag{4}$$

Given the output $\mathbf{v} = [\mathbf{v}_1, \mathbf{v}_2]$, these expressions are trivially invertible:

$$\mathbf{u}_2 = (\mathbf{v}_2 - t_1(\mathbf{v}_1)) \odot \exp(-s_1(\mathbf{v}_1)), \qquad \mathbf{u}_1 = (\mathbf{v}_1 - t_2(\mathbf{u}_2)) \odot \exp(-s_2(\mathbf{u}_2)). \tag{5}$$

Importantly, the mappings $s_i$ and $t_i$ can be arbitrarily complicated functions of $\mathbf{v}_1$ and $\mathbf{u}_2$ and need *not* themselves be invertible. In our implementation, they are realized by a succession of several fully connected layers with leaky ReLU activations.

A deep invertible network is composed of a sequence of these reversible blocks. To increase model capacity, we apply a few simple extensions to this basic architecture. Firstly, if the

dimension $D$ is small, but a complex transformation has to be learned, we find it advantageous to pad both the in- and output of the network with an equal number of zeros. This does not change the intrinsic dimensions of in- and output, but enables the network's interior layers to embed the data into a larger representation space in a more flexible manner. Secondly, we insert permutation layers between reversible blocks, which shuffle the elements of the subsequent layer's input in a randomized, but fixed, way. This causes the splits $\mathbf{u} = [\mathbf{u}_1, \mathbf{u}_2]$ to vary between layers and enhances interaction among the individual variables. Kingma and Dhariwal (2018) use a similar architecture with learned permutations.

### 3.3 Bi-directional training

Invertible networks offer the opportunity to simultaneously optimize for losses on both the in- and output domains (Grover et al., 2017), which allows for more effective training. Hereby, we perform forward and backward iterations in an alternating fashion, accumulating gradients from both directions before performing a parameter update. For the forward iteration, we penalize deviations between simulation outcomes $\mathbf{y}_i = s(\mathbf{x}_i)$ and network predictions $f_{\mathbf{y}}(\mathbf{x}_i)$ with a loss $\mathcal{L}_{\mathbf{y}}\big(\mathbf{y}_i, f_{\mathbf{y}}(\mathbf{x}_i)\big)$. Depending on the problem, $\mathcal{L}_{\mathbf{y}}$ can be any supervised loss, e.g. squared loss for regression or cross-entropy for classification.

The loss for latent variables penalizes the mismatch between the joint distribution of network outputs $q\big(\mathbf{y} = f_{\mathbf{y}}(\mathbf{x}), \mathbf{z} = f_{\mathbf{z}}(\mathbf{x})\big) = p(\mathbf{x})/|J_{\mathbf{yz}}|$ and the product of marginal distributions of simulation outcomes $p\big(\mathbf{y} = s(\mathbf{x})\big) = p(\mathbf{x})/|J_s|$ and latents $p(\mathbf{z})$ as $\mathcal{L}_{\mathbf{z}}\big(q(\mathbf{y}, \mathbf{z}), p(\mathbf{y})\,p(\mathbf{z})\big)$.

We block the gradients of $\mathcal{L}_{\mathbf{z}}$ with respect to $\mathbf{y}$ to ensure the resulting updates only affect the predictions of $\mathbf{z}$ and do not worsen the predictions of $\mathbf{y}$. Thus, $\mathcal{L}_{\mathbf{z}}$ enforces two things: firstly, the generated $\mathbf{z}$ must follow the desired normal distribution $p(\mathbf{z})$; secondly, $\mathbf{y}$ and $\mathbf{z}$ must be independent upon convergence (i.e. $p(\mathbf{z}\,|\,\mathbf{y}) = p(\mathbf{z})$), and not encode the same information twice. As $\mathcal{L}_{\mathbf{z}}$ is implemented by Maximum Mean Discrepancy $D$ (Sec. 3.4), which only requires samples from the distributions to be compared, the Jacobian determinants $J_{\mathbf{yz}}$ and $J_s$ do not have to be known explicitly. In appendix Sec. 1, we prove the following theorem:

**Theorem:** *If an INN* $f(\mathbf{x}) = [\mathbf{y}, \mathbf{z}]$ *is trained as proposed, and both the supervised loss* $\mathcal{L}_{\mathbf{y}} = \mathbb{E}[(\mathbf{y} - f_{\mathbf{y}}(\mathbf{x}))^2]$ *and the unsupervised loss* $\mathcal{L}_{\mathbf{z}} = D\big(q(\mathbf{y}, \mathbf{z}), p(\mathbf{y})\,p(\mathbf{z})\big)$ *reach zero, sampling according to Eq. 1 with* $g = f^{-1}$ *returns the true posterior* $p(\mathbf{x}\,|\,\mathbf{y}^*)$ *for any measurement* $\mathbf{y}^*$.

Although $\mathcal{L}_{\mathbf{y}}$ and $\mathcal{L}_{\mathbf{z}}$ are sufficient asymptotically, a small amount of residual dependency between $\mathbf{y}$ and $\mathbf{z}$ remains after a finite amount of training. This causes $q(\mathbf{x}\,|\,\mathbf{y})$ to deviate from the true posterior $p(\mathbf{x}\,|\,\mathbf{y})$. To speed up convergence, we also define a loss $\mathcal{L}_{\mathbf{x}}$ on the input side, implemented again by MMD. It matches the distribution of backward predictions $q(\mathbf{x}) = p\big(\mathbf{y} = f_{\mathbf{y}}(\mathbf{x})\big)\,p\big(\mathbf{z} = f_{\mathbf{z}}(\mathbf{x})\big)/|J_{\mathbf{x}}|$ against the prior data distribution $p(\mathbf{x})$ through $\mathcal{L}_{\mathbf{x}}\big(p(\mathbf{x}), q(\mathbf{x})\big)$. In the appendix, Sec. 1, we prove that $\mathcal{L}_{\mathbf{x}}$ is guaranteed to be zero when the forward losses $\mathcal{L}_{\mathbf{y}}$ and $\mathcal{L}_{\mathbf{z}}$ have converged to zero. Thus, incorporating $\mathcal{L}_{\mathbf{x}}$ does not alter the optimum, but improves convergence in practice.

Finally, if we use padding on either network side, loss terms are needed to ensure no information is encoded in the additional dimensions. We *a)* use a squared loss to keep those values close to zero and *b)* in an additional inverse training pass, overwrite the padding dimensions with noise of the same amplitude and minimize a reconstruction loss, which forces these dimensions to be ignored.

### 3.4 Maximum mean discrepancy

Maximum Mean Discrepancy (MMD) is a kernel-based method for comparison of two probability distributions that are only accessible through samples (Gretton et al., 2012). While a trainable discriminator loss is often preferred for this task in high-dimensional problems, especially in GAN-based image generation, MMD also works well, is easier to use and much cheaper, and leads to more stable training (Tolstikhin et al., 2017). The method requires a kernel function as a design parameter, and we found that kernels with heavier tails than Gaussian are needed to get meaningful gradients for outliers. We achieved best results with the Inverse Multiquadratic $k(\mathbf{x}, \mathbf{x}') = 1/(1 + \|(\mathbf{x} - \mathbf{x}')/h\|_2^2)$, reconfirming the

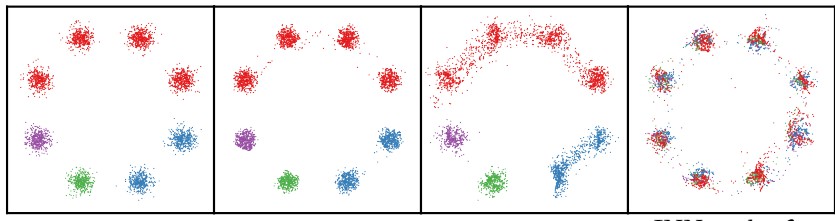

Ground truth    INN, all losses  INN, only $\mathcal{L}_\mathbf{y} + \mathcal{L}_\mathbf{z}$  INN, only $\mathcal{L}_\mathbf{x}$

Figure 2: **Viability of INN for a basic inverse problem.** The task is to produce the correct (multi-modal) distribution of 2D points $\mathbf{x}$, given only the color label $\mathbf{y}^*$. When trained with all loss terms from Sec. 3.3, the INN output matches ground truth almost exactly *(2nd image)*. The ablations *(3rd and 4th image)* show that we need $\mathcal{L}_\mathbf{y}$ and $\mathcal{L}_\mathbf{z}$ to learn the conditioning correctly, whereas $\mathcal{L}_\mathbf{x}$ helps us remain faithful to the prior.

suggestion from Tolstikhin et al. (2017). Since the magnitude of the MMD depends on the kernel choice, the relative weights of the losses $\mathcal{L}_\mathbf{x}$, $\mathcal{L}_\mathbf{y}$, $\mathcal{L}_\mathbf{z}$ are adjusted as hyperparameters, such that their effect is about equal.

## 4  EXPERIMENTS

We first demonstrate the capabilities of INNs on two well-behaved synthetic problems and then show results for two real-world applications from the fields of medicine and astrophysics. Additional details on the datasets and network architectures are provided in the appendix.

### 4.1  ARTIFICIAL DATA

**Gaussian mixture model:**    To test basic viability of INNs for inverse problems, we train them on a standard 8-component Gaussian mixture model $p(\mathbf{x})$. The forward process is very simple: The first four mixture components (clockwise) are assigned label $\mathbf{y} = \textcolor{red}{\textbf{red}}$, the next two get label $\mathbf{y} = \textcolor{blue}{\textbf{blue}}$, and the final two are labeled $\mathbf{y} = \textcolor{green}{\textbf{green}}$ and $\mathbf{y} = \textcolor{purple}{\textbf{purple}}$ (Fig. 2). The true inverse posteriors $p(\mathbf{x} \,|\, \mathbf{y}^*)$ consist of the mixture components corresponding to the given one-hot-encoded label $\mathbf{y}^*$. We train the INN to directly regress one-hot vectors $\mathbf{y}$ using a squared loss $\mathcal{L}_\mathbf{y}$, so that we can provide plain one-hot vectors $\mathbf{y}^*$ to the inverse network when sampling $p(\mathbf{x} \,|\, \mathbf{y}^*)$.  We observe the following:  (i) The INN learns very accurate approximations of the posteriors and does not suffer from mode collapse. (ii) The coupling block architecture does not reduce the network's representational power – results are similar to standard networks of comparable size (see appendix Sec. 2). (iii) Bidirectional training works best, whereas forward training alone (using only $\mathcal{L}_\mathbf{y}$ and $\mathcal{L}_\mathbf{z}$) captures the conditional relationships properly, but places too much mass in unpopulated regions of $\mathbf{x}$-space. Conversely, pure inverse training (just $\mathcal{L}_\mathbf{x}$) learns the correct $\mathbf{x}$-distribution, but loses all conditioning information.

**Inverse kinematics:** For a task with a more complex and continuous forward process, we simulate a simple inverse kinematics problem in 2D space: An articulated arm moves vertically along a rail and rotates at three joints. These four degrees of freedom constitute the parameters $\mathbf{x}$. Their priors are given by a normal distribution, which favors a pose with $180°$ angles and centered origin. The forward process is to calculate the coordinates of the end point $\mathbf{y}$, given a configuration $\mathbf{x}$. The inverse problem asks for the posterior distribution over all possible inputs $\mathbf{x}$ that place the arm's end point at a given $\mathbf{y}$ position. An example for a fixed $\mathbf{y}^*$ is shown in Fig. 3, where we compare our INN to a conditional VAE (see appendix Fig. 7 for conceptual comparison of architectures). Adding Inverse Autoregressive Flow (IAF, Kingma et al., 2016) does not improve cVAE performance in this case (see appendix, Table 2). The $\mathbf{y}^*$ chosen in Fig. 3 is a hard example, as it is unlikely under the prior $p(\mathbf{x})$ (Fig. 3, *right*) and has a strongly bi-modal posterior $p(\mathbf{x} \,|\, \mathbf{y}^*)$.

In this case, due to the computationally cheap forward process, we can use approximate Bayesian computation (ABC, see appendix Sec. 7) to sample from the ground truth posterior. Compared to ground truth, we find that both INN and cVAE recover the two symmetric

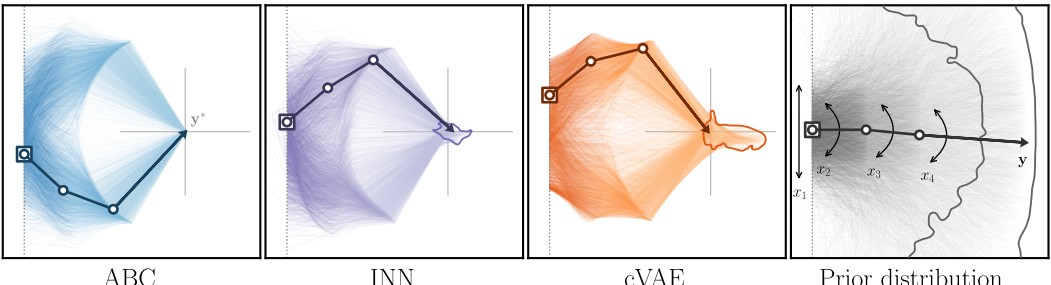

Figure 3: **Distribution over articulated poses x, conditioned on the end point $\mathbf{y}^*$.**
The desired end point $\mathbf{y}^*$ is marked by a gray cross. A dotted line on the left represents the
rail the arm is based on, and the faint colored lines indicate sampled arm configurations $\mathbf{x}$
taken from the true (ABC) or learned (INN, cVAE) posterior $p(\mathbf{x} \,|\, \mathbf{y}^*)$. The prior *(right)* is
shown for reference. The actual end point of each sample may deviate slightly from the target
$\mathbf{y}^*$; contour lines enclose the regions containing 97% of these end points. We emphasize the
articulated arm with the highest estimated likelihood for illustrative purposes.

modes well. However, the true end points of $\mathbf{x}$-samples produced by the cVAE tend to miss
the target $\mathbf{y}^*$ by a wider margin. This is because the forward process $\mathbf{x} \rightarrow \mathbf{y}$ is only learned
implicitly during cVAE training. See appendix for quantitative analysis and details.

## 4.2  REAL-WORLD APPLICATIONS

After demonstrating the viability on synthetic data, we apply our method to two real world
problems from medicine and astronomy. While we focus on the medical task in the following,
the astronomy application is shown in Fig. 5.

In medical science, the functional state of biological tissue is of interest for many applications.
Tumors, for example, are expected to show changes in oxygen saturation $s_{O_2}$ (Hanahan and
Weinberg, 2011). Such changes cannot be measured directly, but influence the reflectance of
the tissue, which can be measured by multispectral cameras (Lu and Fei, 2014). Since ground
truth data can not be obtained from living tissue, we create training data by simulating
observed spectra $\mathbf{y}$ from a tissue model $\mathbf{x}$ involving $s_{O_2}$, blood volume fraction $v_{\text{hb}}$, scattering
magnitude $a_{\text{mie}}$, anisotropy $g$ and tissue layer thickness $d$ (Wirkert et al., 2016). This model
constitutes the forward process, and traditional methods to learn point estimates of the
inverse (Wirkert et al., 2016; 2017; Claridge and Hidovic-Rowe, 2013) are already sufficiently
reliable to be used in clinical trials. However, these methods can not adequately express
uncertainty and ambiguity, which may be vital for an accurate diagnosis.

**Competitors.**  We train an INN for this problem, along with two ablations (as in Fig. 2),
as well as a cVAE with and without IAF (Kingma et al., 2016) and a network using the
method of Kendall and Gal (2017), with dropout sampling and additional aleatoric error
terms for each parameter. The latter also provides a point-estimate baseline (classical NN)
when used without dropout and error terms, which matches the current state-of-the-art
results in Wirkert et al. (2017). Finally, we compare to ABC, approximating $p(\mathbf{x} \,|\, \mathbf{y}^*)$ with
the 256 samples closest to $\mathbf{y}^*$. Note that with enough samples ABC would produce the true
posterior. We performed 50 000 simulations to generate samples for ABC at test time, taking
one week on a GPU, but still measure inconsistencies in the posteriors. The learning-based
methods are trained within minutes, on a training set of 15 000 samples generated offline.

**Error measures.**  We are interested in both the accuracy (point estimates), and the shape
of the posterior distributions. For point estimates $\hat{\mathbf{x}}$, i.e. MAP estimates, we compute the
deviation from ground-truth values $\mathbf{x}^*$ in terms of the RMSE over test set observations $\mathbf{y}^*$,
RMSE $= \sqrt{\mathbb{E}_{\mathbf{y}^*}[\|\hat{\mathbf{x}} - \mathbf{x}^*\|^2]}$. The scores are reported both for the main parameter of interest
$s_{O_2}$, and the parameter subspace of $s_{O_2}$, $v_{\text{hb}}$, $a_{\text{mie}}$, which we found to be the only recoverable
parameters. Furthermore, we check the re-simulation error: We apply the simulation $s(\hat{\mathbf{x}})$ to
the point estimate, and compare the simulation outcome to the conditioning $\mathbf{y}^*$. To evaluate
the shape of the posteriors, we compute the calibration error for the sampling-based methods,
based on the fraction of ground truth inliers $\alpha_{\text{inl.}}$ for corresponding $\alpha$-confidence-region of

Table 1: **Quantitative results in medical application.** We measure the accuracy of point/MAP estimates as detailed in Sec. 4.2. Best results within measurement error are **bold**, and we determine uncertainties ($\pm$) by statistical bootstrapping. The parameter $s_{O_2}$ is the most relevant in this application, whereas *error all* means all recoverable parameters ($s_{O_2}$, $v_{hb}$ and $a_{mie}$). Re-simulation error measures how well the MAP estimate $\hat{x}$ is conditioned on the observation $y^*$. Calibration error is the most important, as it summarizes correctness of the posterior shape in one number; see appendix Fig. 11 for more calibration results.

| Method | MAP error $s_{O_2}$ | MAP error all | MAP re-simulation error | **Calibration error** |
|---|---|---|---|---|
| NN (+ Dropout) | $0.057 \pm 0.003$ | $\mathbf{0.56 \pm 0.01}$ | $0.397 \pm 0.008$ | $1.91\%$ |
| INN | $\mathbf{0.041 \pm 0.002}$ | $\mathbf{0.57 \pm 0.02}$ | $\mathbf{0.327 \pm 0.007}$ | $\mathbf{0.34\%}$ |
| INN, only $\mathcal{L}_y, \mathcal{L}_z$ | $0.066 \pm 0.003$ | $0.71 \pm 0.02$ | $0.506 \pm 0.010$ | $1.62\%$ |
| INN, only $\mathcal{L}_x$ | $0.861 \pm 0.033$ | $1.70 \pm 0.02$ | $2.281 \pm 0.045$ | $3.20\%$ |
| cVAE | $0.050 \pm 0.002$ | $0.74 \pm 0.02$ | $\mathbf{0.314 \pm 0.007}$ | $2.19\%$ |
| cVAE-IAF | $0.050 \pm 0.002$ | $0.74 \pm 0.03$ | $\mathbf{0.313 \pm 0.008}$ | $1.40\%$ |
| ABC | *$0.036 \pm 0.001$* | *$0.54 \pm 0.02$* | *$0.284 \pm 0.005$* | *$0.90\%$* |
| Simulation noise | | | $0.129 \pm 0.001$ | |

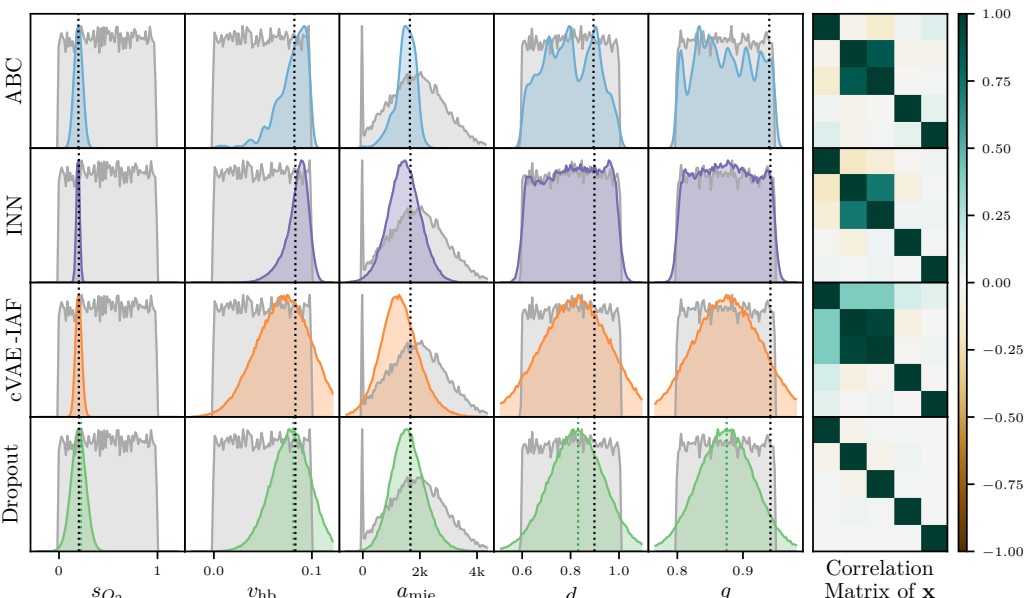

Figure 4: **Sampled posterior of 5 parameters for fixed $y^*$ in medical application.** For a fixed observation $y^*$, we compare the estimated posteriors $p(x \,|\, y^*)$ of different methods. The bottom row also includes the point estimate *(dashed green line)*. Ground truth values $x^*$ *(dashed black line)* and prior $p(x)$ over all data *(gray area)* are provided for reference.

the marginal posteriors of $x$. The reported error is the median of $|\alpha_{inl.} - \alpha|$ over all $\alpha$. All values are computed over 5000 test-set observations $y^*$, or 1000 observations in the case of re-simulation error. Each posterior uses 4096 samples, or 256 for ABC; all MAP estimates are found using the mean-shift algorithm.

**Quantitative results.** Evaluation results for all methods are presented in Table 1. The INN matches or outperforms other methods in terms of point estimate error. Its accuracy deteriorates slightly when trained without $\mathcal{L}_x$, and entirely when trained without the conditioning losses $\mathcal{L}_y$ and $\mathcal{L}_z$, just as in Fig. 2. For our purpose, the calibration error is the most important metric, as it summarizes the correctness of the whole posterior distribution in one number (see appendix Fig. 11). Here, the INN has a big lead over cVAE(-IAF) and Dropout, and even over ABC due to the low ABC sample count.

**Qualitative results.** Fig. 4 shows generated parameter distributions for one fixed measurement $y^*$, comparing the INN to cVAE-IAF, Dropout sampling and ABC. The three former methods use a sample count of 160 000 to produce smooth curves. Due to the sparse posteri-

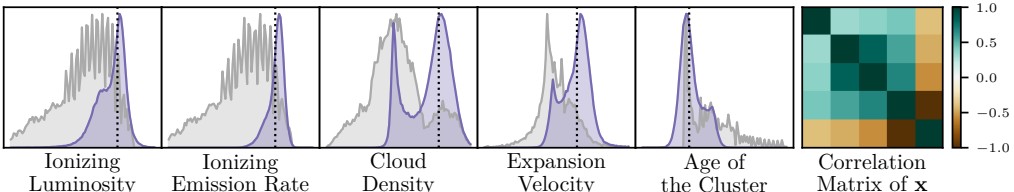

| Ionizing Luminosity | Ionizing Emission Rate | Cloud Density | Expansion Velocity | Age of the Cluster | Correlation Matrix of **x** |

Figure 5: **Astrophysics application.** Properties **x** of star clusters in interstellar gas clouds are inferred from multispectral measurements **y**. We train an INN on simulated data, and show the sampled posterior of 5 parameters for one $\mathbf{y}^*$ (colors as in Fig. 4, second row). The peculiar shape of the prior is due to the dynamic nature of these simulations. We include this application as a real-world example for the INN's ability to recover multiple posterior modes, and strong correlations in $p(\mathbf{x} \,|\, \mathbf{y}^*)$, see details in appendix, Sec. 5.

ors of 256 samples in the case of ABC, kernel density estimation was applied to its results, with a bandwidth of $\sigma = 0.1$. The results produced by the INN provide relevant insights: First, we find that the posteriors for layer thickness $d$ and anisotropy $g$ match the shape of their priors, i.e. $\mathbf{y}^*$ holds no information about these parameters – they are unrecoverable. This finding is supported by the ABC results, whereas the other two methods misleadingly suggest a roughly Gaussian posterior. Second, we find that the sampled distributions for the blood volume fraction $v_{\mathrm{hb}}$ and scattering amplitude $a_{\mathrm{mie}}$ are strongly correlated (rightmost plot). This phenomenon is not an analysis artifact, but has a sound physical explanation: As blood volume fraction increases, more light is absorbed inside the tissue. For the sensor to record the same intensities $\mathbf{y}^*$ as before, scattering must be increased accordingly. In Fig. 10 in the appendix, we show how the INN is applied to real multispectral images.

## 5 CONCLUSION

We have shown that the full posterior of an inverse problem can be estimated with invertible networks, both theoretically and practically on problems from medicine and astrophysics. We share the excitement of the application experts to develop INNs as a generic tool, helping them to better interpret their data and models, and to improve experimental setups. As a side effect, our results confirm the findings of others that the restriction to coupling layers does not noticeably reduce the expressive power of the network.

In summary, we see the following fundamental advantages of our INN-based method compared to alternative approaches: Firstly, one can learn the forward process and obtain the (more complicated) inverse process 'for free', as opposed to e.g. cGANs, which focus on the inverse and learn the forward process only implicitly. Secondly, the learned posteriors are not restricted to a particular parametric form, in contrast to classical variational methods. Lastly, in comparison to ABC and related Bayesian methods, the generation of the INN posteriors is computationally very cheap. In future work, we plan to systematically analyze the properties of different invertible architectures, as well as more flexible models utilizing cycle losses, in the context of representative inverse problem. We are also interested in how our method can be scaled up to higher dimensionalities, where MMD becomes less effective.

ACKNOWLEDGMENTS

LA received funding by the Federal Ministry of Education and Research of Germany, project 'High Performance Deep Learning Framework' (No 01IH17002). JK, CR and UK received financial support from the European Research Council (ERC) under the European Unions Horizon 2020 research and innovation program (grant agreement No 647769). SW and LMH received funding from the European Research Council (ERC) starting grant COMBIOSCOPY (637960). EWP, DR, and RSK acknowledge support by Collaborative Research Centre (SFB 881) 'The Milky Way System' (subprojects B1, B2 and B8), the Priority Program SPP 1573 'Physics of the Interstellar Medium' (grant numbers KL 1358/18.1, KL 1358/19.2 and GL 668/2-1) and the European Research Council in the ERC Advanced Grant STARLIGHT (project no. 339177)

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

# Appendix

## 1 Proof of correctness of generated posteriors

**Lemma:** *If some bijective function $f : x \to z$ transforms a probability density $p_X(x)$ to $p_Z(z)$, then the inverse function $f^{-1}$ transforms $p_Z(z)$ back to $p_X(x)$.*

**Proof:** We denote the probability density obtained through the reverse transformation as $p_X^*(x)$. Therefore, we have to show that $p_X^*(x) = p_X(x)$. For the forward direction, via the change-of-variables formula, we have

$$p_Z(z) = p_X\Big(x = f^{-1}(z)\Big) \left|\det[\partial_z(f^{-1})]\right| \tag{6}$$

with the Jacobian $\partial_z f^{-1} \equiv \partial f_i^{-1}/\partial z_j$. For the reverse transformation, we have

$$p_X^*(x) = p_Z\Big(z = f(x)\Big) \left|\det[\partial_x f]\right| . \tag{7}$$

We can substitute $p_Z$ from Eq. 6 and obtain

$$p_X^*(x) = p_X\Big(x = f^{-1}(f(x))\Big) \left|\det\left[(\partial_z(f^{-1}))(\partial_x f)\right]\right| \tag{8}$$

$$= p_X(x) \left|\det\left[(\partial_z f^{-1})(\partial_x f)\right]\right| \tag{9}$$

$$= p_X(x) \left|\det[I]\right| = p_X(x). \tag{10}$$

In Eq. 9, the Jacobians cancel out due to the inverse function theorem, i.e. the Jacobian $\partial_z(f^{-1})$ is the matrix inverse of $\partial_x f$.

**Theorem:** *If an INN $f(\mathbf{x}) = [\mathbf{y}, \mathbf{z}]$ is trained as proposed, and both the supervised loss $\mathcal{L}_\mathbf{y} = \mathbb{E}[(\mathbf{y} - f_\mathbf{y}(\mathbf{x}))^2]$ and the unsupervised loss $\mathcal{L}_\mathbf{z} = D\big(q(\mathbf{y}, \mathbf{z}), p(\mathbf{y})\,p(\mathbf{z})\big)$ reach zero, sampling according to Eq. 1 with $g = f^{-1}$ returns the true posterior $p(\mathbf{x} \,|\, \mathbf{y}^*)$ for any measurement $\mathbf{y}^*$.*

**Proof:** We denote the chosen latent distribution as $p_Z(\mathbf{z})$, the distribution of observations as $p_Y(\mathbf{y})$, and the joint distribution of network outputs as $q(\mathbf{y}, \mathbf{z})$. As shown by Gretton et al. (2012), if the MMD loss converges to 0, the network outputs follow the prescribed distribution:

$$\mathcal{L}_\mathbf{z} = 0 \iff q(\mathbf{y}, \mathbf{z}) = p_Y(\mathbf{y})\, p_Z(\mathbf{z}) \tag{11}$$

Suppose we take a posterior conditioned on a fixed $\mathbf{y}^*$, i.e. $p(\mathbf{x} \,|\, \mathbf{y}^*)$, and transform it using the forward pass of our perfectly converged INN. From this we obtain an output distribution $q^*(\mathbf{y}, \mathbf{z})$. Because $\mathcal{L}_\mathbf{y} = 0$, we know that the output distribution of $\mathbf{y}$ (marginalized over $\mathbf{z}$) must be $q^*(\mathbf{y}) = \delta(\mathbf{y} - \mathbf{y}^*)$. Also, because of the independence between $\mathbf{z}$ and $\mathbf{y}$ in the output, the distribution of $\mathbf{z}$-outputs is still $q^*(\mathbf{z}) = p_Z(\mathbf{z})$. So the joint distribution of outputs is

$$q^*(\mathbf{y}, \mathbf{z}) = \delta(\mathbf{y} - \mathbf{y}^*)\, p_Z(\mathbf{z}) \tag{12}$$

When we invert the network, and repeatedly input $\mathbf{y}^*$ while sampling $\mathbf{z} \sim p_Z(\mathbf{z})$, this is the same as sampling $[\mathbf{y}, \mathbf{z}]$ from the $q^*(\mathbf{y}, \mathbf{z})$ above. Using the Lemma from above, we know that the inverted network will output samples from $p(\mathbf{x} \,|\, \mathbf{y}^*)$.

**Corollary:** *If the conditions of the theorem above are fulfilled, the unsupervised reverse loss $\mathcal{L}_\mathbf{x} = D\big(q(\mathbf{x}), p_X(\mathbf{x})\big)$ between the marginalized outputs of the inverted network, $q(\mathbf{x})$, and the prior data distribution, $p_X(\mathbf{x})$, will also be 0. This justifies using the loss on the prior to speed up convergence, without altering the final results.*

**Proof:** Due to the theorem, the estimated posteriors generated by the INN are correct, i.e. $q(\mathbf{x} \mid \mathbf{y}^*) = p(\mathbf{x} \mid \mathbf{y}^*)$. If they are marginalized over observations $\mathbf{y}^*$ from the training data, then $q(\mathbf{x})$ will be equal to $p_X(\mathbf{x})$ by definition. As shown by Gretton et al. (2012), this is equivalent to $\mathcal{L}_{\mathbf{x}} = 0$.

## 2 Artificial data – Gaussian mixture

In Sec. 4.1, we demonstrate that the proposed INN can approximate the true posteriors very well and is not hindered by the required coupling block architecture. Here we show how some existing methods do on the same task, using neural networks of similar size as the INN.

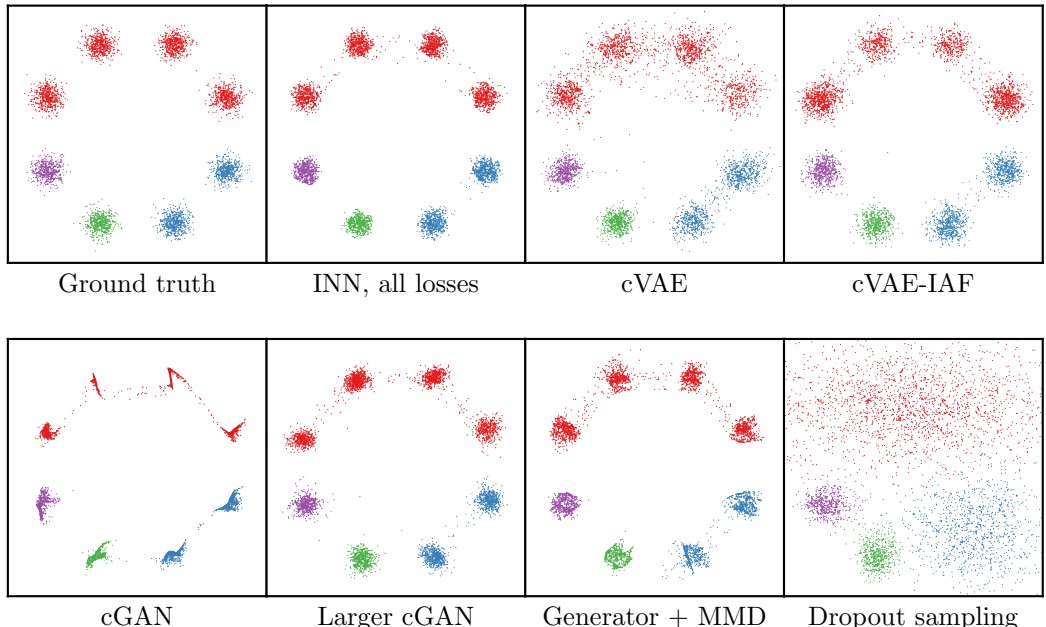

Figure 6: Results of several existing methods for the Gaussian mixture toy example.

**cGAN** Training a conditional GAN of network size comparable to the INN (counting only the generator) and only two noise dimensions turned out to be challenging. Even with additional pre-training to avoid mode collapse, the individual modes belonging to one label are reduced to nearly one-dimensional structures.

**Larger cGAN** In order to match the results of the INN, we trained a more complex cGAN with 2M parameters instead of the previous 10K, and a latent dimension of 128, instead of 2. To prevent mode collapse, we introduced an additional regularization: an extra loss term forces the variance of generator outputs to match the variance of the training data prior. With these changes, the cGAN can be seen to recover the posteriors reasonably well.

**Generator + MMD** Another option is to keep the cGAN generator the same size as our INN, but replace the discriminator with an MMD loss (cf. Sec. 3.4). This loss receives a concatenation of the generator output $\mathbf{x}$ and the label $\mathbf{y}$ it was supplied with, and compares these batch-wise with the concatenation of ground truth $(\mathbf{x}, \mathbf{y})$-pairs. Note that in contrast to this, the corresponding MMD loss of the INN only receives $\mathbf{x}$, and no information about $\mathbf{y}$. For this small toy problem, we find that the hand-crafted MMD loss dramatically improves results compared to the smaller learned discriminator.

**cVAE** We also compare to a conditional Variational Autoencoder of same total size as the INN. There is some similarity between the training setup of our method (Fig. 7, *right*) and

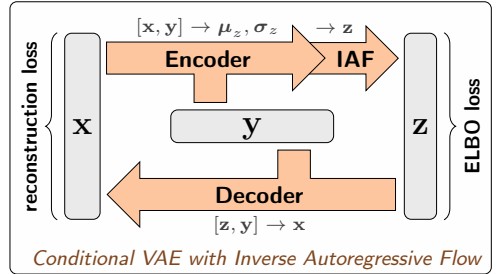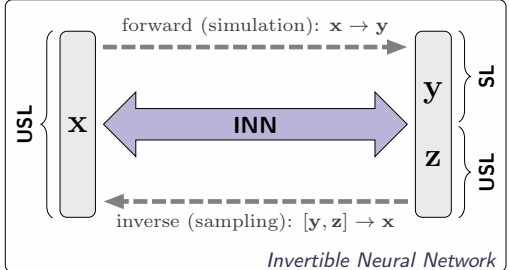

Figure 7: Abstraction of the cVAE-IAF training scheme compared to our INN from Fig. 1. For the standard cVAE, the IAF component is omitted.

that of cVAE (Fig. 7, *left*), as the forward and inverse pass of an INN can also be seen as an encoder-decoder pair. The main differences are that the cVAE learns the relationship $\mathbf{x} \to \mathbf{y}$ only indirectly, since there is no explicit loss for it, and that the INN requires no reconstruction loss, since it is bijective by construction.

**cVAE-IAF** We adapt the cVAE to use Inverse Autoregressive Flow (Kingma et al., 2016) between the encoder and decoder. On the Gaussian mixture toy problem, the trained cVAE-IAF generates correct posteriors on par with our INN (see Fig. 6).

**Dropout sampling** The method of dropout sampling with learned error terms is by construction not able to produce multi-modal outputs, and therefore fails on this task.

## 2.1 LATENT SPACE ANALYSIS

To analyze how the latent space of our INN is structured for this task, we choose a fixed label $\mathbf{y}^*$ and sample $\mathbf{z}$ from a dense grid. For each $\mathbf{z}$, we compute $\mathbf{x}$ through our inverse network and colorize this point in latent ($\mathbf{z}$) space according to the distance from the closest mode in $\mathbf{x}$-space. We can see that our network learns to shape the latent space such that each mode receives the expected fraction of samples (Fig. 8).

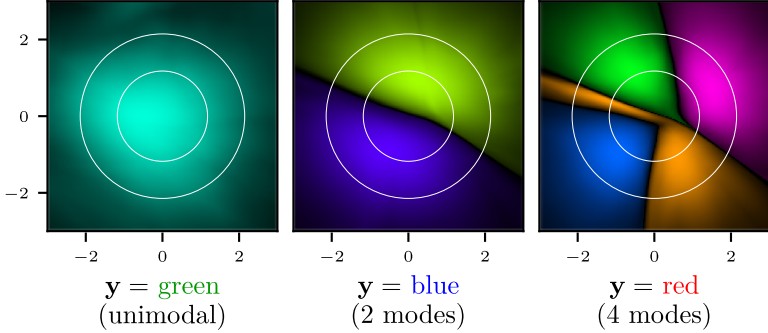

Figure 8: **Layout of INN latent space for one fixed label $\mathbf{y}^*$**, colored by mode closest to $\mathbf{x} = g(\mathbf{y}^*, \mathbf{z})$. For each latent position $\mathbf{z}$, the hue encodes which mode the corresponding $\mathbf{x}$ belongs to and the luminosity encodes how close $\mathbf{x}$ is to this mode. Note that colors used here do not relate to those in Fig. 2, and encode the position $\mathbf{x}$ instead of the label $\mathbf{y}$. The first three columns correspond to labels *green*, *blue* and *red* Fig. 2. White circles mark areas that contain 50% and 90% of the probability mass of latent prior $p(\mathbf{z})$.

## 3 ARTIFICIAL DATA – INVERSE KINEMATICS

A short video demonstrating the structure of our INN's latent space can be found under `https://gfycat.com/SoggyCleanHog`, for a slightly different arm setup.

The dataset is constucted using gaussian priors $x_i \sim \mathcal{N}(0, \sigma_i)$, with $\sigma_1 = 0.25$ and $\sigma_2 = \sigma_3 = \sigma_4 = 0.5 \triangleq 28.65°$. The forward process is given by

$$y_1 = x_1 + l_1 \sin(x_2) + l_2 \sin(x_3 - x_2) + l_3 \sin(x_4 - x_2 - x_3) \tag{13}$$
$$y_2 = l_1 \cos(x_2) + l_2 \cos(x_3 - x_2) + l_3 \cos(x_4 - x_2 - x_3) \tag{14}$$

with the arm lenghts $l_1 = 0.5$, $l_2 = 0.5$, $l_3 = 1.0$.

To judge the quality of posteriors, we quantify both the re-simulation error and the calibration error over the test set, as in Sec. 4.2 of the paper. Because of the cheap simulation, we average the re-simulation error over the whole posterior, and not only the MAP estimate. In Table 2, we find that the INN has a clear advantage in both metrics, confirming the observations from Fig. 3.

Table 2: Quantitative evaluation of the inverse kinematics experiment

| Method | Mean re-sim. err. | Median re-sim. err. | Calibration err. |
|---|---|---|---|
| cVAE | 0.0368 | 0.0307 | 7.78% |
| cVAE-IAF | 0.0368 | 0.0307 | 7.81% |
| INN | 0.0139 | 0.0113 | 0.96% |

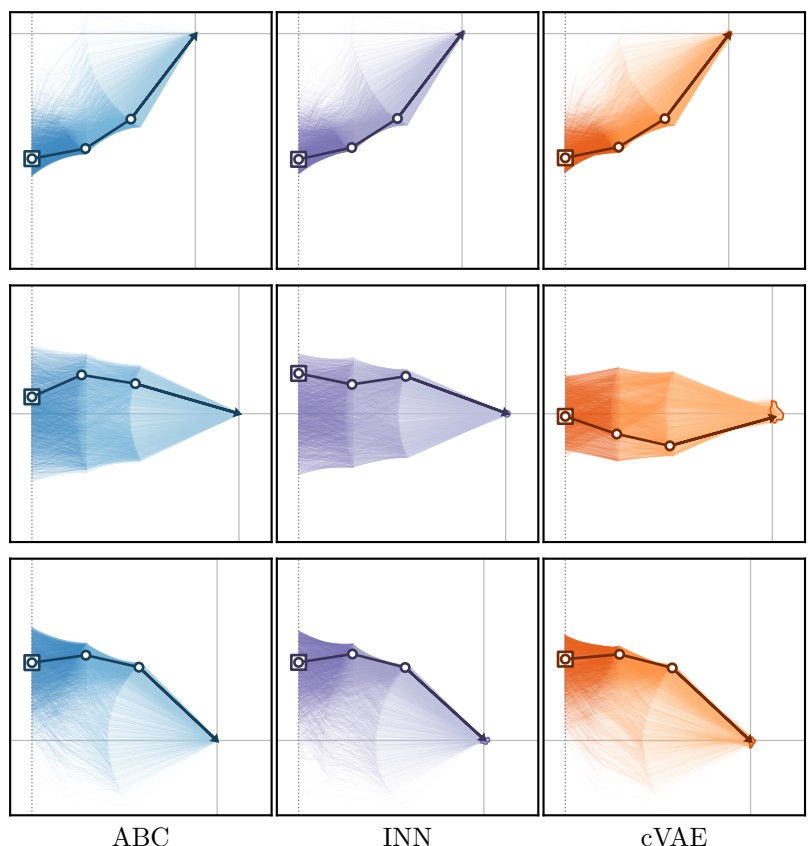

Figure 9: Posteriors generated for less challenging observations $\mathbf{y}^*$ than in Fig. 3.

## 4 MULTISPECTRAL MEASUREMENTS OF BIOLOGICAL TISSUE

The following figure shows the results when the INN trained in Sec. 4.2 is applied pixel-wise to multispectral endoscopic footage. In addition to estimating the oxygenation $s_{O_2}$, we measure the uncertainty in the form of the 68% confidence interval.

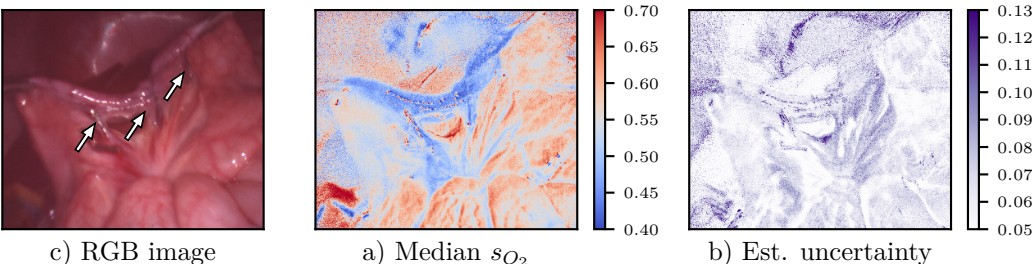

c) RGB image      a) Median $s_{O_2}$      b) Est. uncertainty

Figure 10: **INN applied to real footage to predict oxygenation $s_{O_2}$ and uncertainty.** The clips *(arrows)* on the connecting tissue cause lower oxygenation *(blue)* in the small intestine. Uncertainty is low in crucial areas and high only at some edges and specularities.

## 5 STAR CLUSTER SPECTRAL DATA

Star clusters are born from a large reservoir of gas and dust that permeates the Galaxy, the interstellar medium (ISM). The densest parts of the ISM are called molecular clouds, and star formation occurs in regions that become unstable under their own weight. The process is governed by the complex interplay of competing physical agents such as gravity, turbulence, magnetic fields, and radiation; with stellar feedback playing a decisive regulatory role (S. Klessen and C. O. Glover, 2016). To characterize the impact of the energy and momentum input from young star clusters on the dynamical evolution of the ISM, astronomers frequently study emission lines from chemical elements such as hydrogen or oxygen. These lines are produced when gas is ionized by stellar radiation, and their relative intensities depend on the ionization potential of the chemical species, the spectrum of the ionizing radiation, the gas density as well as the 3D geometry of the cloud, and the absolute intensity of the radiation (Pellegrini et al., 2011). Key diagnostic tools are the so-called BPT diagrams (after Baldwin et al., 1981) emission of ionized hydrogen, H+ , to normalize the recombination lines of O++ , O+ and S+ (see also Kewley et al., 2013). We investigate the dynamical feedback of young star clusters on their parental cloud using the WARPFIELD 1D model developed by Rahner et al. (2017). It follows the entire temporal evolution of the system until the cloud is destroyed, which could take several stellar populations to happen. At each timestep we employ radiative transfer calculations (Reissl et al., 2016) to generate synthetic emission line maps which we use to train the neural network. Similar to the medical application from Section 4.2, the mapping from simulated observations to underlying physical parameters (such as cloud and cluster mass, and total age of the system) is highly degenerate and ill-posed. As an intermediary step, we therefore train our forward model to predict the observable quantities $\mathbf{y}$ (emission line ratios) from composite simulation outputs $\mathbf{x}$ (such as ionizing luminosity and emission rate, cloud density, expansion velocity, and age of the youngest cluster in the system, which in the case of multiple stellar populations could be considerably smaller than the total age). Using the inverse of our trained model for a given set of observations $\mathbf{y}^*$, we can obtain a distribution over the unobservable properties $\mathbf{x}$ of the system.

Results for one specific y are shown in Fig. 5. Note that our network recovers a decidedly multimodal distribution of $\mathbf{x}$ that visibly deviates from the prior $p(\mathbf{x})$. Note also the strong correlations in the system. For example, the measurements $\mathbf{y}^*$ investigated may correspond to a young cluster with large expansion velocity, or to an older system that expands slowly. Finding these ambiguities in $p(\mathbf{x} \,|\, \mathbf{y}^*)$ and identifying degeneracies in the underlying model are pivotal aspects of astrophysical research, and a method to effectively approximate full posterior distributions has the potential to lead to a major breakthrough in this field.

## 6 CALIBRATION CURVE FOR TISSUE PARAMETER ESTIMATION

In Sec. 4.2, we report the median calibration error for each method. The following figure plots the calibration error, $q_{\text{inliers}} - q$, against the level of confidence $q$. Negative values mean that a model is overconfident, while positive values say the opposite.

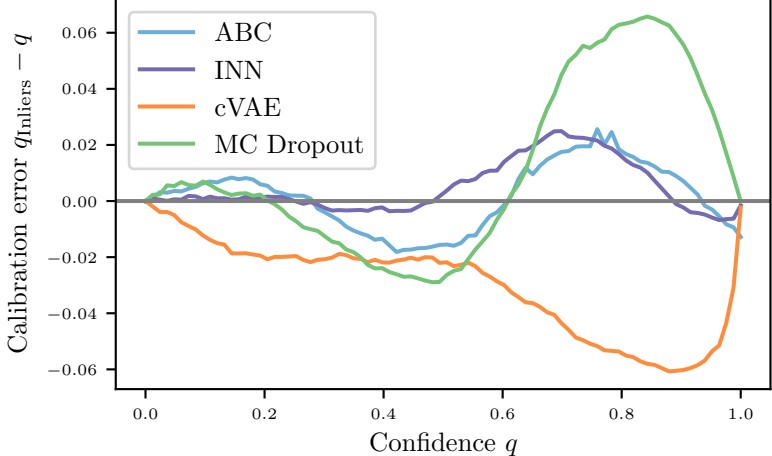

Figure 11: Calibration curves for all four methods compared in Sec. 4.2.

# 7 Approximate Bayesian computation (ABC)

While there is a whole field of research concerned with ABC approaches and their efficiency-accuracy tradeoffs, our use of the method here is limited to the essential principle of rejection sampling. When we require $N$ samples of $\mathbf{x}$ from the posterior $p(\mathbf{x} \,|\, \mathbf{y}^*)$ conditioned on some $\mathbf{y}^*$, there are two basic ways to obtain them:

**Threshold:** We set an acceptance threshold $\epsilon$, repeatedly draw $\mathbf{x}$-samples from the prior, compute the corresponding $\mathbf{y}$-values (via simulation) and keep those where $\mathrm{dist}(\mathbf{y}, \mathbf{y}^*) < \epsilon$, until we have accepted $N$ samples. The smaller we want $\epsilon$, the more simulations have to be run, which is why we use this approach only for the experiment in Sec. 4.1, where we can afford to run the forward process millions or even billions of times.

**Quantile:** Alternatively, we choose what quantile $q$ of samples shall be accepted, and then run exactly $N/q$ simulations. All sampled pairs $(\mathbf{x}, \mathbf{y})$ are sorted by $\mathrm{dist}(\mathbf{y}, \mathbf{y}^*)$ and the $N$ closest to $\mathbf{y}^*$ form the posterior. This allows for a more predictable runtime when the simulations are costly, as in the medical application in Sec. 4.2 where $q = 0.005$.

# 8 Details of datasets and network architectures

Table 3 summarizes the datasets used throughout the paper. The architecture details are given in the following.

Table 3: Dimensionalities and training set sizes for each experiment.

| Experiment | training data | dim($\mathbf{x}$) | dim($\mathbf{y}$) | dim($\mathbf{z}$) | see also |
|---|---|---|---|---|---|
| Gaussian mixture | $10^6$ | 2 | 8 | 2 | |
| Inverse kinematics | $10^6$ | 4 | 2 | 2 | |
| Medical data | 15 000 | 13 | 8 | 13 | Wirkert et al. (2016) |
| Astronomy | 8 772 | 19 | 69 | 17 | Pellegrini et al. (2011) |

## 8.1 Artificial data – Gaussian mixture

**INN:** 3 invertible blocks, 3 fully connected layers per affine coefficient function with ReLU activation functions in the intermediate layers, zero padding to a nominal dimension of 16, Adam optimizer, decaying learning rate from $10^{-3}$ to $10^{-5}$, batch size 200. The inverse multiquadratic kernel was used for MMD, with $h = 0.2$ in both $\mathbf{x}$- and $\mathbf{z}$-space.

**Dropout sampling:** 6 fully connected layers with ReLU activations, Adam optimizer, learning rate decay from $10^{-3}$ to $10^{-5}$, batch size 200, dropout probability $p = 0.2$.

**cGAN:** 6 fully connected layers for the generator and 8 for the discriminator, all with leaky ReLU activations. Adam was used for the generator, SGD for the discriminator, learning rates decaying from $2 \cdot 10^{-3}$ to $2 \cdot 10^{-6}$, batch size 256. Initially 100 iterations training with $\mathcal{L} = \frac{1}{N} \sum_i \|g(z_i, y_i) - x_i\|_2^2$, to separate the differently labeled modes, followed by pure GAN training.

**Larger cGAN:** 2 fully connected layers with 1024 neurons each for discriminator and generator, batch size 512, Adam optimizer with learning rate $8 \cdot 10^{-4}$ for the generator, SGD with learning rate $1.2 \cdot 10^{-3}$ and momentum 0.05 for the discriminator, $1.6 \cdot 10^{-3}$ weight decay for both, 0.25 dropout probabiliy for the generator at training and test time. Equal weighting of discriminator loss and penalty of output variance $\mathcal{L} = (\text{Var}_i[g(z_i, y_i)] - \text{Var}_i[x_i])^2$

**Generator with MMD:** 8 fully connected layers with leaky ReLU activations, Adam optimizer, decaying learning rate from $10^{-3}$ to $10^{-6}$, batch size 256. Inverse multiquadratic kernel, $h = 0.5$.

**cVAE:** 3 fully connected layers each for encoder and decoder, ReLU activations, learning rate $2 \cdot 10^{-2}$, decay to $2.5 \cdot 10^{-5}$, Adam optimizer, batch size 25, reconstruction loss weighted 50:1 versus KL divergence loss.

## 8.2    ARTIFICIAL DATA – INVERSE KINEMATICS

**INN:** 6 affine coupling blocks with 3 fully connected layers each and leaky ReLU activations. Adam optimizer, decaying learning rate from $10^{-2}$ to $10^{-4}$, multiquadratic kernel with $h = 1.2$.

**cVAE:** 4 fully connected layers each for encoder and decoder, ReLU activations, learning rate $5 \cdot 10^{-3}$, decay to $1.6 \cdot 10^{-5}$, Adam optimizer, batch size 250, reconstruction loss weighted 15:1 versus KL divergence loss.

## 8.3    FUNCTIONAL PARAMETER ESTIMATION FROM MULTISPECTRAL TISSUE IMAGES

**INN:** 3 invertible blocks, 4 fully connected layers per affine coefficient function with leaky ReLUs in the intermediate layers, zero padding to double the original width. Adam optimizer, learning rate decay from $2 \cdot 10^{-3}$ to $2 \cdot 10^{-5}$, batch size 200. Inverse multiquadratic kernel with $h = 1$, weighted MMD terms by observation distance with decaying $\gamma = 0.2$ to 0.

**Dropout sampling/point estimate:** 8 fully connected layers, ReLU activations, Adam with decaying learning rate from $10^{-2}$ to $10^{-5}$, batch size 100, dropout probability $p = 0.2$.

**cVAE:** 4 fully connected layers each for encoder and decoder, ReLU activations, learning rate $10^{-3}$, decay to $3.2 \cdot 10^{-6}$, Adam optimizer, batch size 25, reconstruction loss weighted $10^3$:1 versus KL divergence loss.

## 8.4    IMPACT OF STAR CLUSTERS ON THE DYNAMICAL EVOLUTION OF THE GALACTIC GAS

**INN:** 5 invertible blocks, 4 fully connected layers per affine coefficient function with leaky ReLUs in the intermediate layers, no additional zero padding. Adam optimizer with decaying learning rate from $2 \cdot 10^{-3}$ to $1.5 \cdot 10^{-6}$, batch size 500. Kernel for latent space: $k(\mathbf{z}, \mathbf{z}') = \exp(-\|(\mathbf{z} - \mathbf{z}')/h\|_2)$ with $h = 7.1$. Kernel for $\mathbf{x}$-space: $k(\mathbf{x}, \mathbf{x}') = -\|\mathbf{x} - \mathbf{x}'\|_{1/2}^{1/4}$. Due to the complex nature of the prior distributions, this was the kernel found to capture the details correctly, whereas the peak of the inverse multiquadratic kernel was too broad for this purpose.

