# OpenReview forum: "Analyzing Inverse Problems with Invertible Neural Networks"
_ICLR.cc/2019/Conference_

### Official Review · AnonReviewer2 · 2018-11-02
**Constraining models to enable approximate posterior inference**

**Rating:** 7
**Confidence:** 2

**Review:**

The authors propose in this paper an approach for learning models with tractable approximate posterior inference. The paper is well motivated (fast and accurate posterior inference) and the construction of the solutions (invertible architecture, appending vectors to input and output, choice of cost function) well described. From my understanding, it seems this method is also to be compatible with other methods of approximate Bayesian Computation (ABC).
Concerning the experimental section:
- The Mixture of Gaussians experiment is a good illustration of how the choice of cost functions influences the solution. However, I do not understand how are the *discrete* output y is handled. Is it indeed a discrete output (problem with lack of differentiability)? Softwax probability? Other modelling choice?
- The inverse kinematics is an interesting illustration of the potential advantage of this method over conditional VAE and how close it is to ABC which can be reasonably computed for this problem.
- For the medical application, INN outperforms other methods (except sometimes for ABC, which is far more expensive, or direct predictor, which doesn’t provide uncertainty estimates) over some metrics such as the error on parameters recovery (Table 1), calibration error, and does indeed have a approximate posterior which seems to correspond to the ABC solution better. I’m not sure I understand what we are supposed to learn from the astrophysics experiments.
The method proposed and the general problem it aims at tackling seem interesting enough, the toy experiments demonstrates well the advantage of the method. However, the real-world experiments are not necessarily the easiest to read.
EDIT: the concerns were mostly addressed in the revision.

---

> ### Author Response · Authors · 2018-11-10
> **Re: Constraining models to enable approximate posterior inference**
>
> Thank you very much for your time, and your constructive comments, we are looking forward to further discussions!
> We answer your questions and concerns in the following.
>
> > "However, I do not understand how are the *discrete* output y is handled."
>
> For this toy problem, we represent labels y by standard one-hot encoding, and we directly regress one-hot vectors using squared loss instead of softmax. This allows us to input one-hot vectors into the inverted network to generate conditional x-samples.
>
> > "I’m not sure I understand what we are supposed to learn from the astrophysics experiments."
>
> We included this experiment to demonstrate that we are able to find multi-modal posteriors in a second real-world setting relevant to natural science.
>
> > "INN outperforms other methods [...] over some metrics such as the error on parameters recovery (Table 1), calibration error, and does indeed have a approximate posterior which seems to correspond to the ABC solution better"
>
> We indeed consider the calibration errors (reported in Sec. 4.2 (“Quantitative results”) and Appendix Sec. 6) the most meaningful of these comparisons, because they directly measure the quality of the estimated posterior distributions, and INNs have a clear lead here.
> We will add these numbers to Table 1 to emphasize their importance.
>
> > "However, the real-world experiments are not necessarily the easiest to read."
>
> We understand, although we tried our best to condense the complicated nature of these applications. For the astrophysics setting, we provide more information in the appendix, Sec. 5, and for the medical application we refer to [1] for full details.
>
> [1] Wirkert et al.: Robust near real-time estimation of physiological parameters from megapixel multispectral images with inverse monte carlo and random forest regression. International Journal of Computer Assisted Radiology and Surgery, 2016.
> (https://link.springer.com/article/10.1007/s11548-016-1376-5 )

---

> ### Author Response · Authors · 2018-11-16
> **Revised Version**
>
> We have uploaded a revised version of the paper, thank you again for your suggestions.
> The changes and additions are highlighted in red font for convenience.
> Please also note that by adding these changes, our page count increased by half a page beyond the recommended 8 pages.
> If this presents a problem, we can attempt shorten the paper accordingly.

---

### Official Review · AnonReviewer3 · 2018-11-02
**Invertible network with observations for posterior probability of complex input distributions with a theoretical valid bidirectional training scheme.**

**Rating:** 6
**Confidence:** 3

**Review:**

While the invertible model structure itself is essentially the same as Real-NVP, the use of observation variables in the framework with theoretically sound bidirectional training for safe use of the seemingly naïve inclusion of y (i.e., y and z can be independent). Its abilities to model the posterior distributions of the inputs are supported by both quantitative and qualitative experiments. The demonstration on practical examples is a plus.

The advantage of INN, however, is not crystal clear to me versus other generative methods such as GAN and VAE. This is an interesting paper overall, so I am looking forward for further discussions.

Pros:
1.	Extensive analyses of the possibility of modeling posterior distributions with an INN have been shown. Detailed experiment setups are provided in the appendix.

2.	The theoretical guarantee (with some assumptions) of the true posterior might be beneficial in practice for relatively low-dimensional or less complex tasks.

Comments/Questions:
1.	From the generative model point of view, could the authors elaborate on the comparison against cGAN (aside from the descriptions in Appendix 2)? It is quoted “cGAN…often lack satisfactory diversity in practice”. Also, can cGAN be used estimate the density of X (posterior or not)?

2.	For the bidirectional training, did the ratios of the losses (L_z, L_y, L_x) have to be changed, or the iterations of forward/backward trainings have to be changed (e.g., 1 forward, 1 backward vs. 2 forward, 1 backward)? This question comes from my observation that the nature of the losses, especially for L_y vs. L_y,L_x (i.e., SL vs. USL) seem to be different.

3.	“we find it advantageous to pad both the in- and output of the network with equal number of zeros”: Is this to effectively increase the intermediate network dimensions? Also, does this imply that for both forward and inverse process those zero-padded entries always come out to be zero? It seems that there needs some way to enforce them to be zero to ensure that the propagation happens only among the entries belonging to the variables of interests (x, y and z).

4.	It seems that most of the experiments are done in relatively small dimensional data. This is not necessarily a drawback, I am curious if this model could succeed on higher dimensional data (e.g., image), especially with the observation y.

---

> ### Author Response · Authors · 2018-11-10
> **Re: Invertible network with observations for posterior probability of complex input distributions with a theoretical valid bidirectional training scheme.**
>
> Thank you very much for your time, and your constructive comments, we are looking forward to further discussions!
> We answer your questions and concerns in the following.
>
> > "The advantage of INN is not crystal clear to me versus other generative methods such as GAN and VAE."
>
> It is indeed possible to adapt other network types to the task of predicting conditional posteriors. We are currently setting up experiments for detailed analysis of the respective advantages and disadvantages and will report about these results in a future paper. In the present paper, we focus on demonstrating that high-quality posteriors can actually be learned using bi-directional training as facilitated by INNs.
>
> Concerning the comments/questions:
> 1.
> > "could the authors elaborate on the comparison against cGAN"
>
> cGAN generators are at an inherent disadvantage relative to INNs, because they never see ground-truth pairs (x,y) directly -- they are only informed about them indirectly via discriminator gradients. This it not a problem for simple relationships, e.g. between images x and attributes y, and cGANs work very well there. However, it makes learning of complicated forward processes much harder and may cause the resulting posteriors to be inaccurate. Moreover, INNs are forced to embed every training point x somewhere in the latent space, whereas cGAN generators may fail to allocate latent space for some x, because this is never explicitly penalized. This can lead to mode collapse and insufficient diversity.
>
> > "Can cGAN be used to estimate the density of X (posterior or not)?"
>
> cGANs can in principle do this by choosing a generator architecture with tractable Jacobian (using e.g. coupling layers or autoregressive flow), but we are not aware of published results about this possibility.
>
> 2.
> > "For the bidirectional training, did the ratios of the losses (L_z, L_y, L_x) have to be changed, or the iterations of forward/backward trainings have to be changed (e.g., 1 forward, 1 backward vs. 2 forward, 1 backward)?"
>
> Yes, the weights of the losses are considered as hyperparameters, because the magnitude of MMD-based losses depends on the chosen kernel function. Hyperparameter optimization suggested an up-weighting of MMD-based losses by a factor of 5, to give them approximately equal impact as the supervised loss.
> For the iterations, we accumulated gradients over one forward and one inverse network execution before each parameter update. We also tried alternating parameter updates after each forward and backward pass, which resulted in equal accuracy, but was a bit slower. We did not experiment with other ratios than 1:1.
>
> 3.
> > "Is this to effectively increase the intermediate network dimensions?"
>
> This is precisely the reason: It improves the representational power of the INN, as mentioned in Sec. 3.2 and discussed in our response to reviewer 1.
> At present, we find this is only necessary for the toy problem in Fig. 2.
>
> > "It seems that there needs some way to enforce them to be zero to ensure that the propagation happens only among the entries belonging to the variables of interests (x, y and z)."
>
> This is correct.
> We explicitly prevent information from being hidden in the padding dimensions in the following way:
> A squared loss ensures that the amplitudes are close to zero.
> In an additional inverse training pass, we overwrite the padding dimensions with noise of the same amplitude, and minimize their effect via a reconstruction loss.
> We will add this to the relevant paragraph in the paper.
>
> 4.
> > "I am curious if this model could succeed on higher dimensional data"
>
> Works such as [1, 2, 3] (also cited in our paper) have shown that the coupling layer architecture in general works well with images. These works use maximum likelihood training, i.e. exploit the tractable Jacobians to maximize the likelihood of the data embedding in latent space. To scale-up our approach, we may need to replace MMD loss with maximum likelihood as well, and first experiments with this show promising results, see
> https://i.imgur.com/ft09Pk9.png .
>
> [1] Laurent Dinh, Jascha Sohl-Dickstein, and Samy Bengio. Density estimation using Real NVP. arXiv:1605.08803, 2016.
> [2] Diederik P Kingma and Prafulla Dhariwal.  Glow: Generative flow with invertible 1x1 convolutions. arXiv:1807.03039, 2018
> [3] Schirrmeister, Robin Tibor, et al. "Generative Reversible Networks." arXiv:1806.01610, 2018

---

> ### Author Response · Authors · 2018-11-16
> **Revised Version**
>
> We have uploaded a revised version of the paper, thank you again for your suggestions.
> The changes and additions are highlighted in red font for convenience.
> Please also note that by adding these changes, our page count increased by half a page beyond the recommended 8 pages.
> If this presents a problem, we can attempt shorten the paper accordingly.

---

### Official Review · AnonReviewer1 · 2018-11-02
**An inspiring idea with weaknesses on theoretical and experimental side**

**Rating:** 7
**Confidence:** 5

**Review:**

1) Summary

The authors propose to use invertible networks to solve ambiguous inverse problems. This is done by training one group of Real-NVP output variables supervised while training the other group via maximum likelihood under a Gaussian prior as done in the standard Real-NVP. Further, the authors suggest to not only train the forward model, but also the inverse model with an MMD critic, similar to previous works that used a more flexible GAN critic [1].

2) Clarity

The paper is easy to understand and the main idea is well-motivated.

3) Significance

The main contribution of this work is of conceptual nature and illustrates how invertible networks are a promising framework for many inverse problems. I really like the main idea and think it is inspiring. However, the experiments and technical contributions are rather limited.

Theoretical / ML contribution:

Using an MMD to factorize groups of latent variables is well-known and combining flow-based maximum likelihood training in the forward model with GAN-like objectives in the inverse model has been done before as well.

Experimental contribution:

I am not fully convinced by the experiments.
The inverse kinematics experiment shows that the posterior collapses from large uncertainty to almost a point for the right-most joint. This seems like a negative result to me.
The medical experiment also seems rather limited, because if I understand correctly the tissue data is artificial and the proposed INN only outperforms competitors (despite ABC) on two out of three measurements. Further, the authors should have explained the experimental setup of the tissue experiment better, as it is not a standard task in the field.
In the astronomy experiment figure 4 shows strong correlations between some of the z variables, the authors claim that this is a feature of their method, but I argue that they should not be present if training with the factorial prior was successful. It would be good to show the correlation between y and z variables as well if they show high dependencies, learning was not very successful. Simply eyeballing the shape of the posterior is not enough to conclude independence.

In summary, even though interesting, the significance of the experimental results is hard to judge and I am a bit worried that if the proposed model is making some strange mistakes on artificial toy-data, how well it will perform on challenging realistic problems.

4) Main Concerns

The authors claim that specifying a prior/posterior distribution in density modeling is complicated and typically the chosen distributions are too simplistic. This argument is, of course, valid, but they also have the same problem and specify z to be factorial Gaussian. So the same "hen-and-egg" problem applies here.

The authors also seem to suggest that they are the first to train flow-based models in forward and inverse direction, but this has already been done in the flow-GAN paper [1].

MMD does not easily scale to high-dimensional problems, this is not a problem here as all artificial problems considered are very low-dimensional. But when applying the proposed algorithm in realistic settings, one will likely need extensions of MMD, like used in MMD GANs, which would introduce min/max games on both sides of the network. This will likely be hard to train and constitutes a fundamental limitation of the approach that needs to be discussed.

5) Minor Concerns

- Some basic citations on normalizing flows seem to be missing, e.g. [2,3].
- How does one guarantee that padded regions are actually zero on output when padding input with zeros? Small variance in those dimensions could potentially code important information. Is this considered as part of y or z?
- The authors require the existence of inverse and set this equal to bijectivity, but injectivity would be sufficient.
- The authors mention that z is conditioned on y, but in their notation, the conditional density p(z|y) never shows up explicitly. It should be made clear, that p(z)=p(z|y) is a consequence of their additional MMD penalty and only holds at convergence.

[1] Grover et al., "Flow-GAN: Combining Maximum Likelihood and Adversarial Learning in Generative Models"
[2] Tabak and Turner, "Density estimation by dual ascent of the log-likelihood"
[3] Deco and Brauer, "Nonlinear higher-order statistical decorrelation by volume-conserving neural architectures"

---

> ### Author Response · Authors · 2018-11-10
> **Re: An inspiring idea with weaknesses on theoretical and experimental side**
>
> Thank you very much for your time, and your constructive comments, we are looking forward to further discussions!
> We answer your questions and concerns in the following.
> Note that we split the response into two comments, due to the 5000 character limit.
>
> > "The inverse kinematics experiment shows that the posterior collapses from large uncertainty to almost a point for the right-most joint. This seems like a negative result to me."
>
> This comment made us realize that the description/illustration of experiment 2 may not have been clear enough.
> The rightmost circle marker is not a joint, but the end effector (‘hand’) of the arm.
> The conditioning variable y is the position of this hand.
> Therefore, having the hand located on or near the gray cross is the desired outcome of the experiment, not a failure.
> The thick contour line does not represent the posterior p(x|y), but indicates the re-simulation error: It is the 97%-confidence region of the model’s end-point distribution p(y|y_target) = integral p(y|x) p(x|y_target) dx and should be as small as possible (ideally, a delta(y - y_target) is desired).
> The ABC result (leftmost panel) is essentially the ground truth posterior.
> We will replace Fig. 3 with the following improved illustration, to clarify the setup and show what the arm’s degrees of freedom are:
> https://i.imgur.com/nNMdwPA.png
>
> > "The medical experiment also seems rather limited, because if I understand correctly the tissue data is artificial and the proposed INN only outperforms competitors (despite ABC) on two out of three measurements. "
>
> Concerning the artificial nature of the medical experiment:
> Medical researchers must resort to simulation, because so far there is no way to create real training data from living tissue.
> These simulations are sufficiently realistic that they are currently used in clinical trials during actual surgery, albeit only with point estimate methods.
> The medical scientists consider our approach a major leap forward, because our full posteriors allow them to quantify uncertainty reliably and efficiently for the first time, especially regarding possible ambiguities arising from multi-modal posteriors.
>
> Concerning the performance measures:
> To compare posteriors, the calibration errors reported in Sec. 4.2 (“Quantitative results”) and Appendix Sec. 6 are the most meaningful performance metrics, and the INN has a clear lead here.
> We will add these numbers to Table 1 to emphasize their importance.
> The numbers in the current Table 1 refer to MAP estimate accuracy, where alternative methods may be competitive, even if their estimated posteriors or uncertainties are inferior.
>
> > "In the astronomy experiment figure 4 shows strong correlations between some of the z variables, the authors claim that this is a feature of their method, but I argue that they should not be present if training with the factorial prior was successful. It would be good to show the correlation between y and z variables as well if they show high dependencies, learning was not very successful."
>
> There seems to be a misunderstanding, the paper does not show the correlation matrix of the latent z variables.
> Instead, the matrices in Figs. 4 and 5 (right) show the correlation of the x-variables for some fixed y.
> It is a distinguishing feature of our method that we can uncover correlations in the posterior p(x|y), which are not visible in the marginals p(x_i|y) or a mean-field approximation.
> We verify correctness of the correlations in Fig. 4 via comparison to (expensive) ABC.
>
> > "The authors also seem to suggest that they are the first to train flow-based models in forward and inverse direction, but this has already been done in the flow-GAN paper [1]. "
>
> Thank you for pointing out that their ‘hybrid’ strategy is equivalent to bi-directional training. We will change the related work and Sec. 3.3, to properly appreciate their pioneering contributions. Note that we did not make any claims to be the first to use bi-directional training.

---

> > ### Author Response · Authors · 2018-11-10
> > **Re: An inspiring idea with weaknesses on theoretical and experimental side (Part 2)**
> >
> > > "The authors claim that specifying a prior/posterior distribution in density modeling is complicated and typically the chosen distributions are too simplistic. This argument is, of course, valid, but they also have the same problem and specify z to be factorial Gaussian. So the same "hen-and-egg" problem applies here."
> >
> > We respectfully disagree with this statement.
> > We only argue that restrictions to the posterior p(x|y) are problematic. In contrast, restricting the latent distribution p(z) to a Gaussian poses no serious limitation, thanks to a theorem in [1]: This paper proves under mild assumptions that any distribution over vectors u can be nonlinearly transformed into a distribution over vectors v, whose elements v_i are independently uniformly distributed in [0,1]^m (“nonlinear independent component analysis”).
> > The uniform distribution can easily be transformed to a Gaussian (or any other desired prior) with standard transformations.
> > Therefore, as long as the neural network is powerful enough and assumptions are fulfilled, it can always realize the transformation from Gaussian p(z) to any arbitrary p(x|y) at any desired accuracy.
> > Note that these properties are not specific to our INN setup, but apply to all models of “normalizing flow”-type.
> >
> > [1] A. Hyvärinen and P. Pajunen. Nonlinear Independent Component Analysis: Existence and Uniqueness results. Neural Networks 12(3): 429--439, 1999.
> > (https://www.cs.helsinki.fi/u/ahyvarin/papers/NN99.pdf )
> >
> > > "MMD does not easily scale to high-dimensional problems, this is not a problem here as all artificial problems considered are very low-dimensional. But when applying the proposed algorithm in realistic settings, one will likely need extensions of MMD, like used in MMD GANs, which would introduce min/max games on both sides of the network."
> >
> > Our paper intentionally includes two real-world examples in order to demonstrate that there are plenty of low-dimensional applications, which will directly profit from our MMD-based solution.
> > Scaling MMD to high dimensions is indeed not easy, and other losses (maximum likelihood, adversarial) may be superior.
> > The following figure shows preliminary results of a forthcoming paper on this subject, where we train using maximum likelihood in conjunction with a supervised classification loss, to enable conditional generation by INNs:
> > https://i.imgur.com/ft09Pk9.png
> >
> > > "- Some basic citations on normalizing flows seem to be missing, e.g. [2,3]."
> >
> > Thank you for pointing these out. It is fascinating to see that some key ideas were already invented 25 years ago. We will add these references.
> >
> > > "- How does one guarantee that padded regions are actually zero on output when padding input with zeros? Small variance in those dimensions could potentially code important information. Is this considered as part of y or z? "
> >
> > We explicitly prevent information from being hidden in the padding dimensions in the following way:
> > * A squared loss ensures that the amplitudes are close to zero.
> > * In an additional inverse training pass, we overwrite the padding dimensions with noise of the same amplitude, and minimize their effect via a reconstruction loss.
> > Note that zero padding of the input is only necessary for the toy problem in Fig. 2, because the width of the resulting network would be too small otherwise.
> > We consider the padding part of y, as it has a supervised loss.
> > We will add this to the relevant paragraph in the paper.
> >
> > > "- The authors require the existence of inverse and set this equal to bijectivity, but injectivity would be sufficient."
> >
> > We think that bijectivity is required for bi-directional training to be well-defined.
> > Since the coupling architecture is bijective by construction, the distinction has no practical implications for our method.
> >
> > > "- The authors mention that z is conditioned on y, but in their notation, the conditional density p(z|y) never shows up explicitly. It should be made clear, that p(z)=p(z|y) is a consequence of their additional MMD penalty and only holds at convergence."
> >
> > You are right, we will make this clear in our revised text.
> >
> > > "[...] I am a bit worried that if the proposed model is making some strange mistakes on artificial toy-data, how well it will perform on challenging realistic problems."
> >
> > We feel that this statement might be due to the misunderstandings discussed in the answers above.
> > There is no indication, quantitatively or otherwise, that our model is behaving incorrectly or unexpectedly in any of the experiments.
> > If this does not answer your concerns, we will be happy to provide further clarifications and additional data.

---

> > > ### Comment · AnonReviewer1 · 2018-11-11
> > > **Thank you for the detailed answers**
> > >
> > > Thank you for clarifying some misconceptions from my side w.r.t. the astronomy experiment and the resulting misinterpretation of your statements about the posterior distribution.
> > >
> > > I was, in fact, referring to practical limitations, related to insufficient expressiveness of the model, that may not make it powerful enough to transform arbitrary densities into the factorized normal space. This is also why a better baseline would be a model that is closer to state-of-the-art density models, like IAF or other normalizing flow extensions to vanilla VAEs.
> > >
> > > In summary, you are trying to improve conditional density estimation and it is not clear why your proposed method should be the method of choice for this if not compared properly to other state-of-the-art conditional density estimation approaches.
> > >
> > > Can you please provide your perspective on this and would you be able to add an additional experiment with a more competitive baseline?
> > >
> > > It would also be great if you upload a revision to incorporate all the changes you mentioned, so I can better judge the current state and clarity of the manuscript.

---

> > > > ### Author Response · Authors · 2018-11-14
> > > > **IAF Baseline**
> > > >
> > > > As you suggested, we incorporated the IAF from [1] into our cVAE. That is, we inserted the IAF subnetwork between the existing encoder and decoder, but didn’t use the more complex decoder from [1], as it did not improve results and destabilized the training.
> > > > Introducing IAF improved results measurably over plain cVAE, at the cost of a larger network. Now, the performance is on par with the INN on the 8 Gaussian mode experiment, but a noticeable gap remains for the inverse kinematics and medical experiments.
> > > > Qualitatively, cVAE-IAF exhibits the same shortcomings as the cVAE, but with reduced magnitude.
> > > >
> > > > The measurements from Table 1 for the cVAE-IAF model are as follows:
> > > > Calibration error: 1.40%
> > > > MAP error s_O2: 0.050 ± 0.002
> > > > MAP error all: 0.74 ± 0.03
> > > > MAP resimulation error: 0.313 ± 0.008
> > > >
> > > > Sampled posteriors for each experiment, comparing INN, cVAE and cVAE-IAF:
> > > > https://i.imgur.com/s2PECtl.jpg
> > > >
> > > > We will upload the revised paper later this week.
> > > >
> > > > Contrary to intuitive expectations, we (and others) found, that the expressive power of INNs relative to unconstrained networks of comparable size, is not substantially reduced. Differences are subtle, and looking at single experiments in isolation may be misleading.
> > > >
> > > > Definitive statements should be based on systematic comparisons along various degrees of freedom:
> > > > - INNs (trained bi-directionally) vs. auto-encoders (trained for cycle consistency), each with several subtypes and network sizes
> > > > - different unsupervised losses (adversarial, MMD, maximum likelihood, information theoretical)
> > > > - different applications and problem sizes
> > > >
> > > > Ideally, the experiments should include more traditional Bayesian methods for the prediction of posteriors as well, e.g. accelerated MCMC and Stein point sampling.
> > > > It will also be interesting to investigate if novel training or prediction schemes enabled by the INNs’ tractable Jacobians can compensate for their potentially reduced expressive power.
> > > >
> > > > We are currently setting up such experiments and will report about our findings in a future paper. In the present paper, we would like to keep the focus on demonstrating that high-quality posteriors can be learned with bi-directional training as facilitated by INNs.
> > > >
> > > > [1] Kingma, Diederik P., et al. "Improved variational inference with inverse autoregressive flow." Advances in Neural Information Processing Systems. 2016.

---

> > > > > ### Comment · AnonReviewer1 · 2018-11-21
> > > > > **Question**
> > > > >
> > > > > Thank you very much for adding the IAF baseline.
> > > > >
> > > > > I have another question about the scope of the method. Can you elaborate on the following and related statements, it seems you suggest there is no inherent advantage of your method compared to related approaches:
> > > > >
> > > > > "Contrary to intuitive expectations, we (and others) found, that the expressive power of INNs relative to unconstrained networks of comparable size, is not substantially reduced. Differences are subtle, and looking at single experiments in isolation may be misleading. ..."
> > > > >
> > > > > Can you please either confirm or explain in what sense the proposed INNs have a fundamental theoretical advantage over competing conditional generative models with respect to learning high quality (i.e. asymptotically correct) posteriors?

---

> > > > > > ### Author Response · Authors · 2018-11-23
> > > > > > **Advantages of INNs**
> > > > > >
> > > > > >
> > > > > > > “it seems you suggest there is no inherent advantage of your method compared to related approaches”
> > > > > >
> > > > > > Our previous comment, “Differences [between INNs and unrestricted architectures] are subtle”, does not refer to INNs applied to inverse problems, but only to the differences in expressive power between these architectures in general.
> > > > > > We were hereby explicitly addressing your comment “I was, in fact, referring to practical limitations, related to insufficient expressiveness of the model”.
> > > > > >
> > > > > > > “Can you please either confirm or explain in what sense the proposed INNs have a fundamental theoretical advantage over competing conditional generative models with respect to learning high quality (i.e. asymptotically correct) posteriors?”
> > > > > >
> > > > > > We see the following fundamental advantages of our INN-based method:
> > > > > > - One can learn the forward process and get the inverse for free (in contrast to e.g. cGAN).
> > > > > > - Posteriors are not restricted to a particular parametric form (in contrast to classical variational methods).
> > > > > > - Posteriors can be efficiently computed (in contrast to e.g. ABC).
> > > > > > - Training converges to the true solution (in contrast to dropout inference).
> > > > > > - One can efficiently calculate the Jacobian of the mapping (which we do not currently take advantage of).
> > > > > > To the best of our knowledge, there are no established approaches with the same properties, beyond the ones discussed in the paper, where INNs are superior.
> > > > > >
> > > > > > Your question whether there are alternative ways to achieve the same goal, and which method works best, is very interesting and will be the focus of another paper after publication of our present results. Our discussions with you helped us identify promising candidates for such comparisons, but we do not consider these alternatives as established methods for our problem setting, so that confident conclusions cannot yet be drawn. We as a community are just starting to learn how to make best use of INNs, and their trade-offs relative to traditional networks need to be investigated further. Overall, all experiments performed to date were highly encouraging.

---

> > > > > > > ### Comment · AnonReviewer1 · 2018-11-23
> > > > > > > **Thanks for the interesting discussion.**
> > > > > > >
> > > > > > > This clarifies most of my points.

---

> > > > > > > > ### Comment · AnonReviewer1 · 2018-11-25
> > > > > > > > **Update**
> > > > > > > >
> > > > > > > > I have raised my rating by one point due to the additional experiments and increased clarity of the revised manuscript.

---

> ### Author Response · Authors · 2018-11-16
> **Revised Version**
>
> We have uploaded a revised version of the paper, thank you again for your suggestions.
> The changes and additions are highlighted in red font for convenience.
> Please also note that by adding these changes, our page count increased by half a page beyond the recommended 8 pages.
> If this presents a problem, we can attempt shorten the paper accordingly.

---

### Public Comment · (anonymous) · 2018-10-21
**Several questions about claims, text clarifications**

Hi,

the work has few interesting extensions to invertible networks. However, there are some questions raised when studying it:

   *  In eq. 3 the authors express the determinant of the jacobian, but it is not clear what partitioning they’ve done to y, z or to x space.

   *  The variable u seems self-defined, what is the relationship with the x, y, z in the previous section?

    * In eq 4, v2 is a function of v1 (which depends on u1) so how come the partial derivative dv2 / du1 = 0? (i.e. how come we end up in a triangular jacobian)

    * The forward and backward iterations (sec. 3.3) are not mentioned in similar works. Could the authors share their experience and or some experimental results of how those help?

    * The authors mention that Lz enforces y and z to be independent. Is there any proof of that? Or did you measure it somehow in the test results?

    * An ablation study justifying all the implementation choices would help. For instance about different architectures of their model, e.g. it seems quite confusing how many invertible blocks are required for similar dimensionality problems. How were those discovered by the authors?

    * Also, the authors mention that Lx contributes marginally, but Table 1 shows that without Lx, the results are worse than all the external compared methods.

---

> ### Author Response · Authors · 2018-10-24
> **Re: Several questions about claims, text clarifications**
>
> Thank you for your interest and your comment. We address your questions in order:
>
> * We are not sure what is meant by this question. We simply concatenate y and z into a single vector, and compute the derivatives with respect to this.
>
> * We use u and v to generically denote the in- and output of each coupling block. For instance, u = x for the first coupling block, and v = [y,z] for the last.
>
> * This is correct. However, as illustrated in the image below, each coupling block consists of two affine transformations. The first of these has an upper triangular Jacobian, and the second has a lower triangular Jacobian. The argument concerning the triangular Jacobians applies to each affine transformation separately. A more in-depth look at the Jacobians of affine coupling layers can be found in Dinh et al. (https://openreview.net/forum?id=HkpbnH9lx Sec. 3.2 and 3.3).
>
> Schematic illustration of coupling block:
> https://i.imgur.com/XdccxeA.png
>
> * As far as we know, we are the first to apply loss functions on both ends of the same network. Our ablations in Fig. 2 and Table 1 show that the method works best when making full use of that. On the practical side, we perform a parameter update once gradients from all loss terms have been accumulated -- an approach also known from GAN training. In our experiments, we found that alternating forward and inverse parameter updates did not affect training results, but increased training time by ~5%.
>
> * L_z is defined as the MMD between the network outputs q(y, z), and the target distribution p(y, z). In our case, y and z are explicitly independent in the target distribution: p(y, z) = p(y)p(z).  When the MMD converges to zero, q is necessarily equal to p, therefore the y and z outputs are asymptotically independent. At present, we do not explicitly differentiate between residual dependency of y and z, and other types of mismatch between the distributions in the case of non-zero loss.
>
> * The network architecture depends on two problem characteristics: Problem dimensionality dictates the width of the layers, and the complexity of the forward process we wish to learn determines the required depth.  We did a coarse grid search to roughly determine the smallest network needed for each application. We will supply ablation studies showing the effect of a larger or smaller number of coupling layers for each of our applications in the following days.
>
> * This is true, the influence of L_x is felt on finite training sets. We meant to say that it plays a smaller role in Table 1 than e.g. in Fig. 2. We will correct our wording in the relevant sections.

---

### Public Comment · ~Robin_Tibor_Schirrmeister1 · 2018-11-07
**Related work using loss at both ends of invertible network**

Hi,

interesting paper.

Just wanted a small reference from our work, that also uses a loss at both ends of the network, albeit only heuristically motivated:
Training generative reversible networks, ICML Workshop on Theoretical Foundations and Applications of Deep Generative Models, https://arxiv.org/abs/1806.01610

Maybe you can find it interesting since you also use a loss at both ends of the network.

Best,
Robin

---

> ### Author Response · Authors · 2018-11-16
> **Revised Version**
>
> We have uploaded a revised version of the paper, and added your paper to the related work section.
> Thank you for the suggestion.

---

> > ### Public Comment · ~Robin_Tibor_Schirrmeister1 · 2018-11-20
> > **Thanks**
> >
> > Great thanks

---

### Meta-Review · Area_Chair1 · 2018-12-13
**Good paper**

**Confidence:** 4
**Recommendation:** Accept (Poster)

**Metareview:**

This paper proposes a framework for using invertible neural networks to study inverse problems, e.g., recover hidden states or parameters of a system from measurements. This is an important and well-motivated topic, and the solution proposed is novel although somewhat incremental. The paper is generally well written. Some theoretical analysis is provided, giving conditions under which the proposed approach recovers the true posterior. Empirically, the approach is tested on synthetic data and real world problems from medicine and astronomy, where it is shown to compared favorably to ABC and conditional VAEs. Adding additional baselines (Bayesian MCMC and Stein methods) would be good. There are some potential issues regarding MMD scalability to high dimensional spaces, but overall the paper makes a solid contribution and all the reviewers agree it should be accepted for publication.